# The genome of opportunistic fungal pathogen *Fusarium oxysporum* carries a unique set of lineage-specific chromosomes

Yong Zhang [1,14], He Yang[1,14], David Turra [2], Shiguo Zhou[3], Dilay Hazal Ayhan[1], Gregory A. DeIulio[1], Li Guo [1,13], Karen Broz [4], Nathan Wiederhold [5], Jeffrey J. Coleman [6], Kerry O' Donnell[7], Ilan Youngster[8], Alexander J. McAdam[9], Sergey Savinov[1], Terrance Shea[10], Sarah Young[10], Qiandong Zeng[10], Martijn Rep[11], Eric Pearlman[12], David C. Schwartz[3], Antonio Di Pietro [2], H. Corby Kistler [4] & Li-Jun Ma [1]*

*Fusarium oxysporum* is a cross-kingdom fungal pathogen that infects plants and humans. Horizontally transferred lineage-specific (LS) chromosomes were reported to determine host-specific pathogenicity among phytopathogenic *F. oxysporum*. However, the existence and functional importance of LS chromosomes among human pathogenic isolates are unknown. Here we report four unique LS chromosomes in a human pathogenic strain NRRL 32931, isolated from a leukemia patient. These LS chromosomes were devoid of housekeeping genes, but were significantly enriched in genes encoding metal ion transporters and cation transporters. Homologs of NRRL 32931 LS genes, including a homolog of ceruloplasmin and the genes that contribute to the expansion of the alkaline pH-responsive transcription factor PacC/Rim1p, were also present in the genome of NRRL 47514, a strain associated with *Fusarium* keratitis outbreak. This study provides the first evidence, to our knowledge, for genomic compartmentalization in two human pathogenic fungal genomes and suggests an important role of LS chromosomes in niche adaptation.

[1] Department of Biochemistry and Molecular Biology, University of Massachusetts Amherst, Amherst, Massachusetts 01003, USA. [2] Departamento de Genetica, Universidad de Cordoba, 14071 Cordoba, Spain. [3] Laboratory for Molecular and Computational Genomics, Laboratory of Genetics and Department of Chemistry, University of Wisconsin-Madison, Madison, Wisconsin 53706, USA. [4] USDA ARS Cereal Disease Laboratory, University of Minnesota, St. Paul, Minnesota 55108, USA. [5] Department of Pathology, University of Texas Health Science Center at San Antonio, San Antonio, Texas 78229, USA. [6] Department of Entomology and Plant Pathology, Auburn University, Auburn, Alabama 36849, USA. [7] National Center for Agricultural Utilization Research, US Department of Agriculture, Agricultural Research Service, Peoria, Illinois 61604, USA. [8] Division of Infectious Diseases, Boston Children's Hospital, Boston, Massachusetts 02115, USA. [9] Department of Laboratory Medicine, Boston Children's Hospital, Boston, Massachusetts 02115, USA. [10] Broad Institute of Harvard and MIT, Cambridge, Massachusetts 02114, USA. [11] Swammerdam Institute for Life Science, University of Amsterdam, Amsterdam, The Netherlands. [12] Institute for Immunology, Physiology and Biophysics, University of California-Irvine, Irvine, California 92697, USA. [13]Present address: School of Electronics and Information Engineering, Xi'an Jiaotong University, Xi'an, Shaanxi Province, China. [14]These authors contributed equally: Yong Zhang, He Yang. *email: lijun@biochem.umass.edu

Each year, fungi infect over 1 billion people and claim around 1.5 million lives worldwide[1]. Advanced medical treatments have increased the complexity of patient populations with immunodeficiency disorders, who are susceptible to opportunistic fungal infections. For instance, chemotherapy increases the survival rate and life expectancy of cancer patients[2] and successful management of immunosuppression prolongs the life expectancy of organ transplant recipients[3]. As a consequence, opportunistic fungi have emerged as an important cause of morbidity and mortality in immunocompromised patients and are posing increasing threats to public health[4,5].

Fusariosis, an invasive fungal infection caused by *Fusarium* spp., is listed as the second most common opportunistic mold infection after aspergillosis[6,7]. Fusarial infections are highly invasive; positive blood cultures were detected among ~50% of fusariosis patients[8]. Infectious keratitis caused by fungal pathogens within the genus *Fusarium* is one of the major causes of corneal infections in the developing world[9,10]. Spread across industrialized countries, *Fusarium* spp. were responsible for fungal keratitis among contact lens wearers, as illustrated by the 2005/06 contact lens-associated *Fusarium* keratitis outbreak[11–14].

As *Fusarium* spp. are broadly resistant to most clinically available antifungals[15], fusariosis in immunocompromised patients is associated with high mortality rates[7] and may approach 100% in persistently neutropenic patients[16,17], and *Fusarium* keratitis is identified as the leading cause of blindness among fungal keratitis patients[18,19].

The ascomycete fungus, *F. oxysporum* constitutes a large species complex that is widely distributed in diverse environments, including soil, indoor environments, and aquatic habitats[4,20,21]. In addition to having multiple clinical manifestations[22], members of the *F. oxysporum* species complex (hereafter FOSC) include common soil-borne plant pathogens that cause devastating vascular wilt diseases[23]. The concept of formae speciales (special forms) was formulated to identify plant pathogens that cause disease on a specific host. However, comparative genome analysis of a tomato (*Solanum lycopersicum*) pathogenic isolate demonstrated that horizontal transfer of lineage-specific (LS) chromosomes can convey pathogenicity on a specific plant host[23,24]. A global survey of the genetic diversity of the FOSC revealed that *F. oxysporum* clinical isolates are phylogenetically diverse[25] and polyphyletic, as previously described for phytopathogenic members of this complex[26]. However, it is unclear whether LS chromosomes are also present in clinical isolates and, if so, whether they contribute to the ability of these fungi to cause fusarioses.

In this study, we analyzed two *F. oxysporum* human isolates, NRRL 32931: (i) a strain isolated from the blood of a leukemia patient with invasive fusariosis; and (ii) NRRL 47514 (MRL 8996), a strain isolated from a contact lens associated with the USA 2005/06 *Fusarium* keratitis outbreak. The genome of NRRL 32931 contained a unique set of four LS chromosomes that were distinct from those previously observed in plant pathogens. Distinct from the phytopathogenic isolates, the human pathogenic isolate NRRL 47514 shared LS sequences with NRRL 32931, including genes involved in metal ion transport and cation transport, genetic traits that could help the pathogen overcome host nutritional immunity and establish mycotic infections. However, the signature effector motif observed in phytopathogenic FOSC genomes[27] was absent in both human isolates. Our results illustrated the potential importance of *F. oxysporum* LS chromosomes for adaptation of the pathogen to the human host.

## Results

***F. oxysporum* human pathogenic isolates**. Two *F. oxysporum* human pathogenic isolates are included in this study.

*F. oxysporum* NRRL 32931 was isolated from a blood culture of a 3-year-old patient being treated with systemic and intrathecal chemotherapy for acute lymphoblastic leukemia, which was diagnosed 6 months earlier. NRRL 47514 was collected from the contaminated contact lens of a patient during the USA 2005/06 *Fusarium* keratitis outbreak[28]. The phylogenetic analysis using 55 conserved single-copy orthologous genes within the *Fusarium* genus confirmed that most human pathogenic isolates are phylogenetically related[22] and two human pathogenic isolates are placed in the same clade, which are within a subclade (100% bootstrap support) comprising isolates Fol4287 (NRRL 34936) of *F. oxysporum* f. sp. *lycopersici*, a vascular wilt pathogen of tomato (*S. lycopersicum*), and Fo47 (NRRL 54002), a nonpathogenic strain used for biological control[29] (Fig. 1 and Supplementary Data 1 and 2).

**Genome of clinical *F. oxysporum* isolate is compartmentalized**. To investigate whether the LS chromosomes reported in phytopathogenic *F. oxysporum* strains are also present in the clinical isolate, we constructed an optical map with over 50-fold physical coverage for the NRRL 32931 genome (Methods), using the restriction enzyme BsiWI. According to the optical map, the NRRL 32931 genome totals 53.42 Mb and contains 15 linkage groups. A comparison with the genome of the tomato pathogenic strain Fol4287[24] revealed 11 homologous chromosomes that represent the core genome and 4 NRRL 32931-specific LS chromosomes, namely chromosomes 12 to 15, which are estimated by optical mapping to be 1.8, 1.3, 1.2, and 1 Mb in size (Supplementary Fig. 1, Fig. 2a), respectively. The presence of four LS chromosomes was confirmed by pulsed-field gel electrophoresis (Fig. 2b), in which the chromosomal sizes were estimated to be 1.8, 1.4, 1.2, and 1.1 Mb, respectively.

The NRRL 32931 genome was sequenced using a whole-genome shotgun approach with Illumina technology (Table 1), with a total of 34,374,476 sequence reads (over 180×). Sequences were assembled using ALLPATHS-LG (Methods)[30]. The assembled genome consists of 168 supercontigs of 47.90 Mb with a supercontig N50 of 4.5 Mb in size. The genome assembly was mapped to the linkage groups defined by optical mapping based on the in silico restriction maps of the assembly (Methods and Supplementary Data 3). The 12 largest supercontigs (a combined size of 44.5 Mb), corresponding to over 93% of the assembled bases (Table 2), were mapped to the 11 chromosomes in the 48 Mb mapping space of the core genome. Over 90% of the genomic spaces defined by the optical map of all core chromosomes were represented in the genome assembly in large supercontigs, reflecting the long-range continuity of the core genome. Due to the high proportion of repeats present, LS chromosomes were highly fragmented. Consequently, mapping of the assembled sequence to the LS chromosomes from 12 to 15 only resulted in 31.1, 42.5, 18.5, and 26.4% mapping rates, respectively. Most sequences belonging to LS chromosomes could not be identified using the optical mapping technique alone, as reliable mapping requires sequences of 50 kb or longer. We thus developed a protocol to eliminate the core genome based on the structural compartmentalization of the genome and conservation of core genomic regions. The protocol identifies core regions, in which more than half of the supercontigs shared over 92% sequence identity with the core of the reference genome Fol4287 (Methods). The remaining supercontigs were classified as NRRL 32931-specific sequences (Supplementary Data 4). This method identified 29 supercontigs as part of the core genome, including all 15 supercontigs identified through optical mapping. The average supercontig size for the core genome was >1.5 Mb. The remaining 139 supercontigs, totaling 3.4 Mb in size (62% of LS

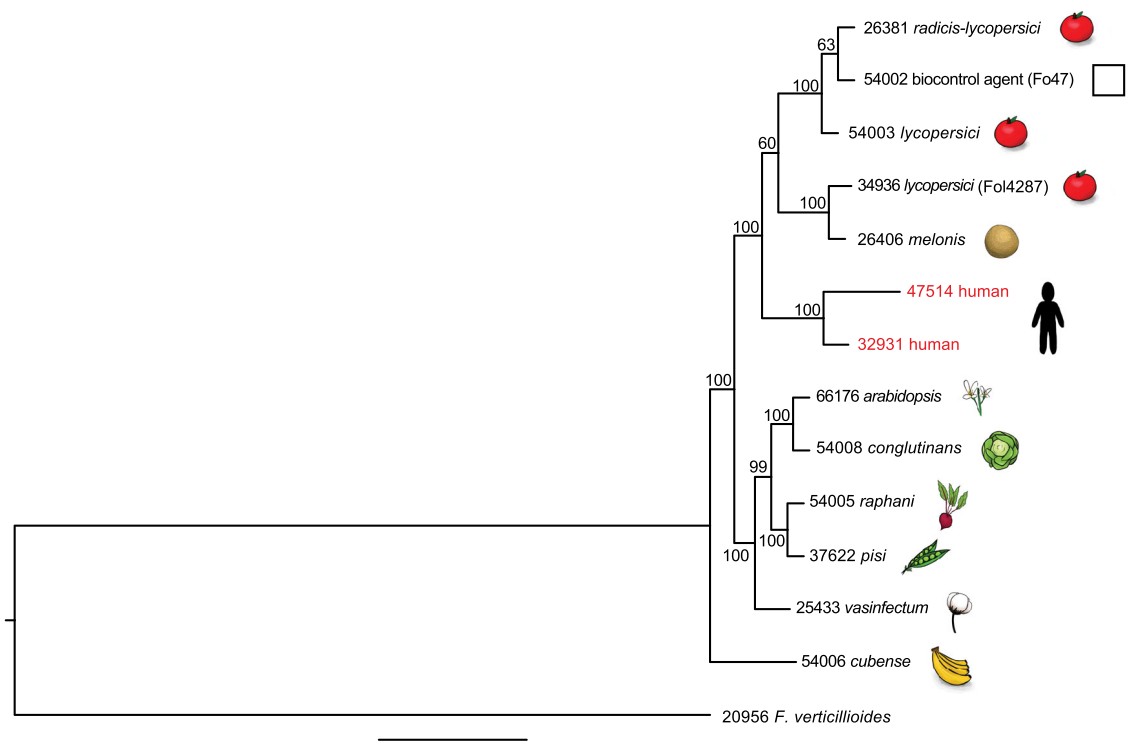

**Fig. 1 Phylogenetic placement of clinical isolate NRRL 32931 within the *F. oxysporum* species complex (FOSC).** The phylogeny is inferred from maximum likelihood analysis of 55 conserved single-copy orthologous genes within the *Fusarium* genus (Supplementary Data 1 and 2), rooted on the sequence of *F. verticillioides*. The bootstrap values, generated through 500 iterations, are indicated above the tree branches. The scale bar represents residue substitution per site. The five-digit strain number represents the accession number in the Agricultural Research Service (ARS) Culture Collection (NRRL), Peoria, IL. Drawings of the hosts/substrates are provided to the right of the NRRL numbers. The biocontrol agent (Fo47) is provided with a box to indicate that it is a non-pathogen.

chromosomes defined by optical mapping), constitute the NRRL 32931 LS genomic regions. The average size of supercontigs of the LS genomic regions was only 24 kb, indicative of severe fragmentation due to the presence of repetitive sequences (Fig. 2c), as previously reported for LS chromosomes in other FOSC genomes[23,24].

The genome of NRRL 32931 encodes 17,280 predicted genes, including 812 (4.7%) LS genes. The gene density of the LS region (1.5 genes per 10 kb window) is about half that of the core (3 genes per 10 kb window). RNA sequencing (RNA-Seq) was employed to assess the completeness of the genome assembly and to assist the genome annotation using the de novo assembler Trinity[31] (details in Methods and Supplementary Data 5 and 6). Over 99.8% of the assembled transcripts (10,105) were aligned to the genome with high confidence, suggesting that our current assembly captured almost all coding sequences.

The keratitis strain, NRRL 47514 (MRL 8996), was sequenced using PacBio and Illumina sequencing (Methods). The assembled genome is 50.11 Mb (252 contigs with an N50 of 1.73 Mb), more than 2 Mb larger than that of the blood strain. This genomic assembly was complete, as it included 99.2% of the fungal genes defined by the Benchmarking Universal Single-Copy Orthologs (BUSCO v3.1). Eleven core chromosomes can be easily mapped to both the phytopathogenic Fol4287 and the blood pathogenic strain NRRL 32931, represented in a total of 38 contigs. LS sequences in the keratitis strain are also repeat rich and are highly fragmented in over 200 contigs (Supplementary Fig. 2). Whereas only 2.3% (56 kb) of the blood strain LS sequences have homologous sequences in the tomato pathogenic strain, more than a third (883 kb) of the blood strain LS sequences have homologous sequences in the keratitis

strain (Fig. 3a and Table 3). Most interestingly, the two human pathogenic strains share almost identical fragments (Fig. 3b). In addition, transposons associated with pathogenicity in some phytopathogenic strains, such as MIMPs[27] and Helitrons[32], were absent from the NRRL 32931 and NRRL 47514 genomes. However, a different subset of transposons, which were mostly characterized as AT-rich repeats, were uniquely present in the NRRL 32931 and NRRL 47514 genomes (Supplementary Data 7).

**Shared LS genes in clinical strains suggest niche adaptation**. LS genes present in both human pathogenic strains were distinct from those in all of the phytopathogenic *F. oxysporum* genomes we have examined to date. For instance, the signature effector genes, *SIX* genes (Secreted In Xylem) and plant cell wall degradation enzymes present in all plant pathogenic *F. oxysporum* genomes[24,33,34] were absent in these two genomes. LS chromosomes in NRRL 32931 were enriched for genes involved in metal ion transport ($p = 2.02 \times 10^{-13}$), cation transport ($p = 3.61 \times 10^{-11}$), and other cellular responses to chemical stimuli through transcription regulation and signal transduction (Supplementary Data 8). Among 812 NRRL 32931 LS genes, the majority (765/812) had homologous hits within the genus of *Fusarium*, reflecting the nature of gene family expansions. We identified 47 LS genes with a potential horizontal origin. Remarkably, 53.2% of those genes (25/47) are present in both the blood strain NRRL 32931 and the keratitis strain NRRL 47514, almost all of which are located in the LS chromosomes (Supplementary Data 9). More than half of these shared horizontally transferred LS genes encode metal ion binding, transport, or response proteins.

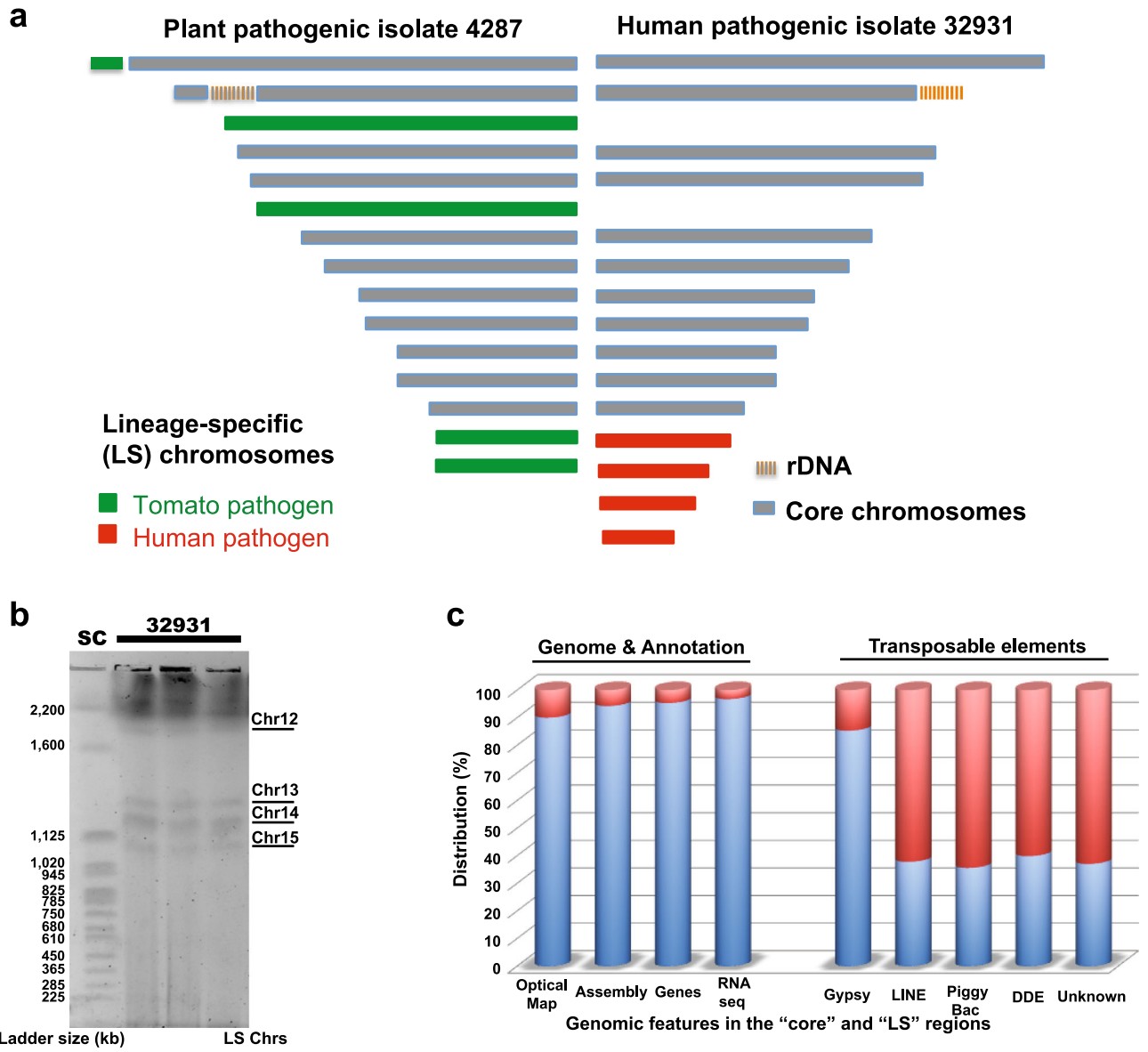

**Fig. 2 Compartmentalization of the NRRL 32931 genome and distribution of genomic features in the core versus LS regions. a** Comparison of optical maps of the human-infecting strain NRRL 32931 and the reference genome Fol4287, a tomato wilt pathogen. The core genome consists of 11 homologous chromosomes (gray) that are conserved within the FOSC[24]. The four unique and smaller lineage-specific chromosomes of human-infecting strain NRRL 32931 are in red. **b** Pulse field gel showing the chromosomal distribution of strain NRRL 32931. All three preps of NRRL 32931 gave the same banding pattern, with four visible LS chromosomes, all below the 2200 kb chromosome band. The estimated sizes of these chromosomes are 1.8 Mb, 1.4 Mb, 1.2 Mb, and 1.1 Mb. SC, *S. cerevisiae* chromosomes. **c** Characterization of the compartmentalized NRRL 32931 genome. Blue shading indicates features in the core and red indicates features in the LS regions. Compared with the core, the LS genome is gene poor and transposon rich.

**Table 1 Genome assembly of NRRL 32931 and NRRL 47514.**

| Strain | Contig statistics | | | | Supercontig statistics | | | | Estimated K-mer and repeat corrected |
|---|---|---|---|---|---|---|---|---|---|
| | Total | Max size (kb)[102] | N50 (kb)[102] | Length (Mb) | Total | Max size (kb)[102] | N50 (kb)[102] | Length (Mb) | Size (Mb)[a] |
| NRRL 32931 | 553 | 2568 | 538 | 47.21 | 168 | 5466 | 4457 | 47.90 | 53.3 |
| NRRL 47514 | 252 | 4647 | 1727 | 50.11 | — | — | — | — | — |

[a]K-mer and repeat-corrected genome size were estimated by establishing the frequency of occurrence of each k-mer (K) size of 96 (the default K size in ALLPATHS-LG) using KmerSpectrum, a module run within ALLPATHS-LG that calculates k-mer frequency statistics on the read data[30].

**Expansion of the *pacC* gene family**. Among the NRRL 32931 LS genes, we observed a striking expansion of the PacC/Rim1p family (Fig. 4a), the members of which encode highly conserved fungal transcription factors that mediate signaling in response to

| Table 2 Mapping the genome assembly of NRRL 32931 to the optical map. | | | |
|---|---|---|---|
| **Linkage groups** | **Map size (Mb)** | **Mapped SCs** | **Mapped SC size (MB)** |
| Chr1 | 6.89 | 12, 4 | 6.7 (97.2%) |
| Chr2 | 5.95 | 1 | 5.8 (97.3%) |
| Chr3 | 5.58 | 98, 2 | 5.4 (96.8%) |
| Chr4 | 5.03 | 3 | 4.9 (96.9%) |
| Chr5 | 4.78 | 5 | 4.6 (96.4%) |
| Chr6 | 4.46 | 47, 6 | 4.3 (96.7%) |
| Chr7 | 3.74 | 7 | 3.5 (94.2%) |
| Chr8 | 3.31 | 8 | 3.1 (94.0%) |
| Chr9 | 3.02 | 21, 9 | 2.8 (93.8%) |
| Chr10 | 2.75 | 10 | 2.6 (92.9%) |
| Chr11 | 2.58 | 11 | 2.4 (94.5%) |
| **Chr12** | **1.80** | **24, 15, 38, 31, 17** | **0.56 (31.1%)** |
| **Chr13** | **1.31** | **18, 28, 13** | **0.56 (42.5%)** |
| **Chr14** | **1.20** | **20, 23, 35** | **0.22 (18.5%)** |
| **Chr15** | **1.02** | **14** | **0.28 (26.4%)** |
| **Total** | **53.42** | | **47.7 (89.3%)** |

Bold font indicates linkage groups identified as NRRL 32931-specific LS chromosomes, whereas the remaining linkage groups represent the core elements of this genome. SC supercontig.

ambient pH[35,36]. In human pathogenic fungi such as *Candida albicans*[37], *Cryptococcus neoformans*[38], *Aspergillus fumigatus*[39], and *F. oxysporum*[40], *pacC* orthologs are essential for full virulence in the mouse model. During pulmonary aspergillosis, PacC governs both breaking through the host physical barrier[41] and adapting to host body conditions[39].

In addition to the full-length *pacC* ortholog (FOYG_02661, *pacC_O*), located on a core chromosome (Chr3), the NRRL 32931 genome encodes three truncated *pacC* homologs, named *pacC_a* (FOYG_15914), *pacC_b* (FOYG_17204), and *pacC_c* (FOYG_17356) (Fig. 4a). The phylogeny suggests an independent origin of all these truncated *pacC* homologs in comparison with the full-length *pacC* ortholog. Even though the DNA-binding zinc finger domain is conserved at the amino acid level in the three additional copies (Fig. 4b), the overall DNA sequence identity with *pacC_O* ranges from 60% for *pacC_b* to 74% for *pacC_a*, which is much lower than that of orthologous *pacC* genes within the FOSC. The presence of all four *pacC* genes in the genome was confirmed using primer sets unique for each gene (Supplementary Fig. 3). Moreover, a 672 bp probe specific for *pacC_b* hybridized to chromosome 14, one of the LS chromosomes (Supplementary Fig. 4), whereas a 952 bp probe specific for the orthologous *pacC_O* hybridized to one of the core chromosomes, as predicted from the genome assembly. Three *pacC* putative paralogs are also present in the keratitis strain. A focused alignment between the supercontig with the *pacC_b* paralog and NRRL 47514 contigs showed that almost the entire supercontig is present in the keratitis strain (Fig. 3b).

The presence of transposable elements was also observed in the flanking regions of the LS *pacC* paralogs (Fig. 4c). For instance,

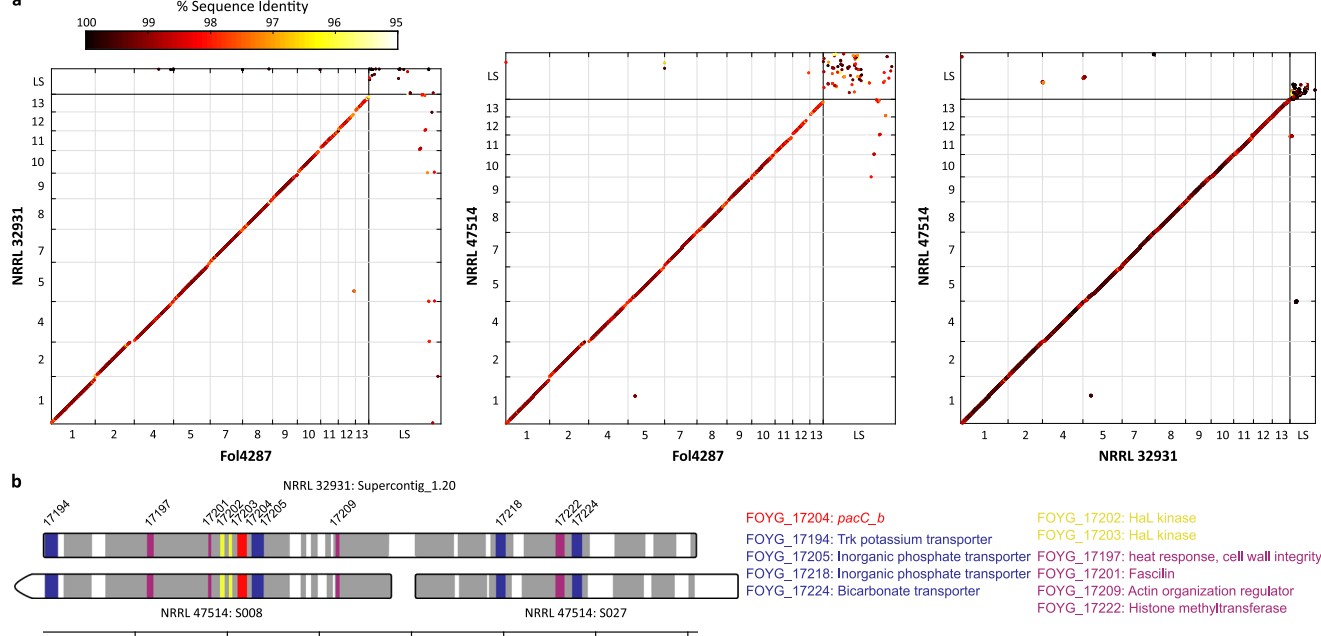

**Fig. 3 Pairwise genome comparisons of NRRL 32931, NRRL 47514, and Fol4287. a** Genome alignments of NRRL 32931 with Fol4287, NRRL 47514 with Fol4287, and NRRL 32931 with NRRL 47514. The LS sequences are combined in one section. In summary, 37% NRRL 32931 masked LS sequences have homologous sequences in the NRRL 47514 genome, whereas there is only 2.3% in the Fol4287 genome. With larger LS region in the NRRL 47514 genome, two human pathogens shared LS sequences and represent 17% of NRRL 47514 masked LS sequences; about 7% of NRRL 47514 masked LS sequences also have homologous sequences in the genome of Fol4287. **b** Alignment of Supercontig 20 of chromosome 14 of NRRL 32931 using ClustalW with default parameters in MEGAX. The supercontig includes *pacC* paralog *pacC_b* to NRRL 47514 LS contigs S008 and S027. Syntenic regions are shown in dark gray and colored bars. Light gray blocks in Supercontig 20 indicate gaps in the sequence. Annotated genes are colored, the gene locus tag numbers are labeled, and more information about the genes is stated below the diagram. Although whole sequences of Supercontig 20 and S027 are shown, the S008 contig is partial. After removing repetitive sequences, the aligned sequences between Supercontig 20 and two NRRL 47514 contigs (S008 and S027) are almost identical, with 99.16% and 98.98% identity across the entire region.

*pacC_a* (FOYG_15914) and an adjacent gene encoding a fungal potassium/sodium efflux P-type ATPase (FOYG_15913) were surrounded by three Gypsy retro-elements, one DNA transposon was located in the *pacC_b* 10 kb flanking region, whereas *pacC_c* was directly flanked by three DNA transposons (Fig. 4c). The nucleotide sequence identity among this expanded *pacC* gene family is below 90%, much lower than that of orthologous genes within the FOSC (~99%), which may result from the horizontal origin of this group of genes or rapid sequence divergence after duplication.

All three additional *pacC* homologs lacked the C-terminus, while containing the intact DNA-binding domain (Fig. 4a). In *Aspergillus nidulans*, and most likely in *F. oxysporum*, PacC is produced as an inactive precursor (>600 aa), which is the predominant form in acidic conditions[42,43]. Upon a shift to neutral or alkaline conditions, the PacC precursor is activated by proteolytic cleavage of ~400 residues from the C-terminus, resulting in a shorter version of the protein (~250 residues)

containing the Zn finger DNA-binding domain, which functions both as an activator of alkaline-expressed genes and as a repressor of acid-expressed genes[44]. Previous studies in *Aspergillus* and *Fusarium* have shown that truncated copies of PacC function as pH-independent dominant activators of alkaline-expressed genes[43,44]. If transcribed, the truncated *pacC* homologs present in strain NRRL 32931 may thus promote fungal adaptation at the slightly alkaline pH of human blood (pH 7.4).

RNA-Seq data from complete medium at pH < 7.0 indicated that the full-length *pacC_O* gene was highly expressed, whereas the truncated *pacC* homologs were expressed to a lesser extent. Quantitative reverse transcription PCR (qRT-PCR) under different pH conditions revealed that *pacC_O* expression was the highest of the *pacC* homologs at all pH values tested, whereas *pacC_b* and *pacC_c* displayed moderate expression, and *pacC_a* had minimal expression. Furthermore, induction of all of these genes, except *pacC_a*, was pH-dependent (Fig. 4d). We confirmed the nuclear localization of both the canonical PacC_O and the truncated PacC_b protein using strains carrying green fluorescent protein-tagged alleles (Fig. 4e).

**Enrichment of proteins with metal ion-binding functions**. One of the most significantly enriched functional categories of NRRL 32931 LS genes was metal ion binding (Supplementary Data 8) related to iron homeostasis, a function reported for many human pathogenic fungi[45,46,47,48]. Among these were three secreted copper-binding proteins with oxidoreductase activities, all located

| Table 3 Length of syntenic sequences of NRRL 32931 with Fol4287 and NRRL 47514. | | | |
|---|---|---|---|
| **Pairwise genome comparison** | **All (bp)** | **Core (bp)** | **LS (bp)** |
| **NRRL 32931 - Fol4287** | 23,527,153 | 23,470,972 | 56,181 |
| **NRRL 32931 - NRRL 47514** | 38,381,056 | 37,497,516 | 883,540 |

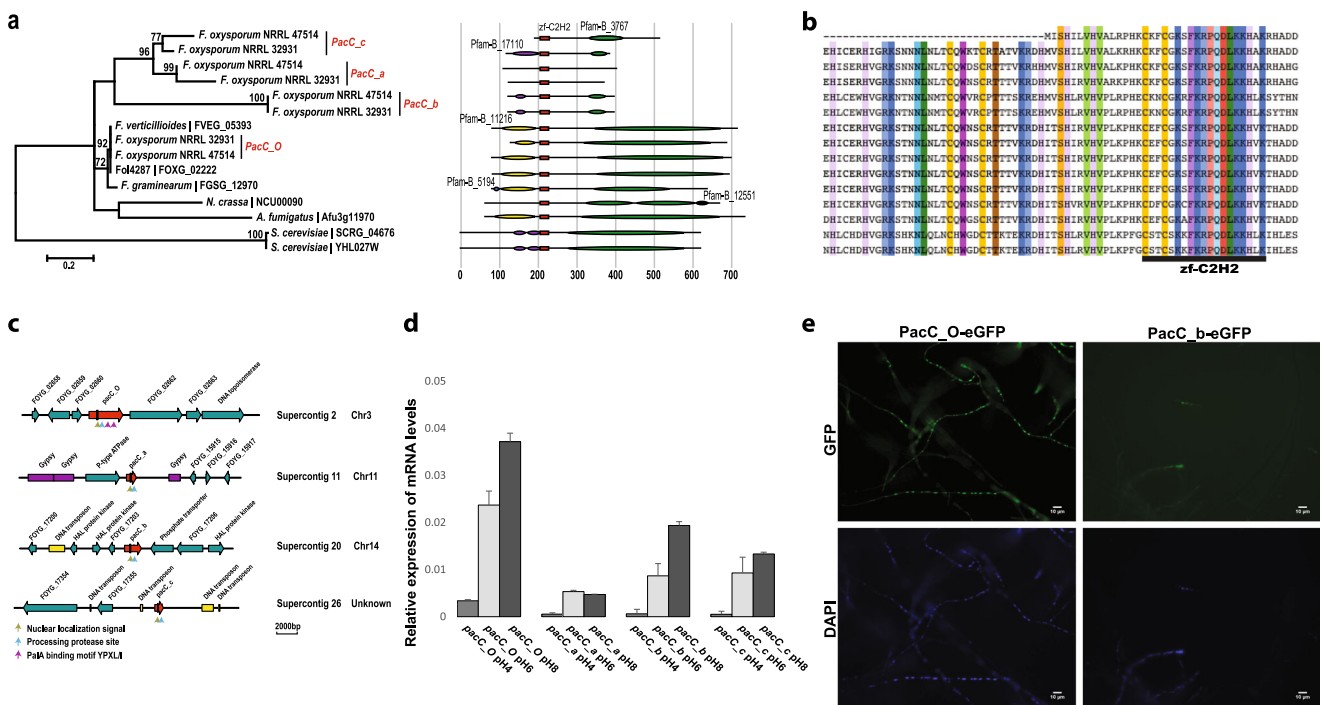

**Fig. 4 Expansion of PacC/Rim1p, a transcription factor mediating adaptation to alkaline pH. a** Phylogeny of *pacC* genes in nine different fungal genomes, including the full-length *pacC* ortholog located on a core chromosome of NRRL 32931 (*pacC_O*, chromosome 3). Bootstrap values above 50 are indicated above the tree branches. Three truncated *pacC* homologs (*pacC_a*, *pacC_b*, and *pacC_c*) encoded in LS regions are uniquely presented in two human pathogenic strains, NRRL 32931 and NRRL 47514. Other fungal *PacC* genes included in this analysis are *F. oxysporum* f. sp. *lycopersici* NRRL 34936 (FOXG_02222), *F. verticillioides* NRRL 20956 (FVEG_05393), *F. graminearum* NRRL 31084 (FGSG_12970), *A. fumigatus* (Afu3g11970), *Neurospora crassa* (NCU00090), *S. cerevisiae* (SCRG_04676), and *S. cerevisiae* (YHL027W). **b** Sequence alignment among *pacC* genes, highlighting conserved amino acids for specific binding to a transcription factor binding site. **c** Truncated *pacC* genes are surrounded by transposable elements, but not the full-length *pacC* ortholog (FOXG_02661). **d** RT-PCR analysis of the different *pacC* genes, showing an induction of expression with a pH increase. Bars represent SD from three biological replicates. **e** Nuclear localization of PacC_O and PacC_b proteins of *F. oxysporum* isolate NRRL 32931. GFP, PacC_O, and PacC_b of *F. oxysporum* isolate NRRL 32931 were tagged with GFP and observed using a confocal microscope; DAPI, nuclei of fungal cells were stained with 4′,6-diamidino-2-phenylindole (DAPI) and observed under a confocal microscope. Scale bar is 10 μm and applies to all images in **e**.

on LS chromosomes and surrounded by transposable elements (FOYG_17133 and FOYG_17127 in chr13 and FOYG_16888 in chr15). For example, the secreted protein FOYG_17127 is a homolog of mammalian ceruloplasmin (CP), the major copper-carrying protein in blood that is essential for modulating copper transport, metal ion homeostasis, and defense against oxidative stress[49]. Intriguingly, no sequences homologous to FOYG_17127 were detected in any plant pathogenic *Fusarium* species. However, single homologs of the gene were identified in other human pathogenic *F. oxysporum* isolates, such as the keratitis strain, which has a homolog sharing 100% identity with FOYG_17127.

Moreover, we identified orthologs in three other opportunistic fungal pathogens, including *Exophiala xenobiotica* (XP_013312618.1) and *Exophiala oligosperma* (XP_016256162.1), which are black yeasts of the Herpotrichiellaceae family that includes diverse human and other vertebrate pathogens[50]; and *Aspergillus calidoustus* (CEL09498.1), a causal agent of invasive aspergillosis[51]. Other than these three human pathogens, only two other fungal genomes, *Penicillium rubens* (XP_002558535.1) and *Acidomyces richmondensis* (KXL41571.1), shared the CP homolog. Both were known to tolerate extreme environmental conditions[52–54]. Apart from fungi, we identified FOYG_17127 homologs with copper-binding properties in 12 bacteria and 5 archaea species (Fig. 5a), most of which were isolated from extreme environments. The presence and phylogenetic status of this homolog in all sequenced organisms provide

strong evidence for the horizontal transmission of FOYG_17127 in human pathogenic *F. oxysporum*.

Human CP is an ancient multicopper oxidase composed of six compact cupredoxin domains containing six tightly bound copper atoms[55]. FOYG_17127 is one-third of the size of the human CP (Fig. 5b), containing two domains that resemble the group 1 and group 2 domains in CP (Supplementary Fig. 5). Most amino acids participating in the active binding of CP to copper are conserved in FOYG_17127 (Fig. 5b). Based on crystal structures of human CP (1KCW)[56] and a homotrimeric complex of a laccase from *Streptomyces* collector (3CG8)[57], we predict that the *F. oxysporum* homolog FOYG_17127 has a homotrimeric quaternary structure with at least four metal atoms per unit (Fig. 5c).

Metals such as zinc, iron, and copper are essential for all living organisms, including infectious microorganisms and their hosts; therefore, metal homeostasis plays an important role at the host–pathogen interface[58]. In humans, nutritional immunity, i.e., controlling the bioavailability of metals by sequestering micronutrients, is used as an active defense mechanism against invading pathogens[59]. To circumvent host defense, fungal pathogens have acquired diverse mechanisms, including a sophisticated iron homeostasis mechanism[60,61]. Human CP carries more than 95% of the total copper in healthy human plasma. The ability of the pathogen to competitively obtain metal from the host could provide an adaptive advantage during infection. Indeed, FOYG_17127 is related to copper-binding

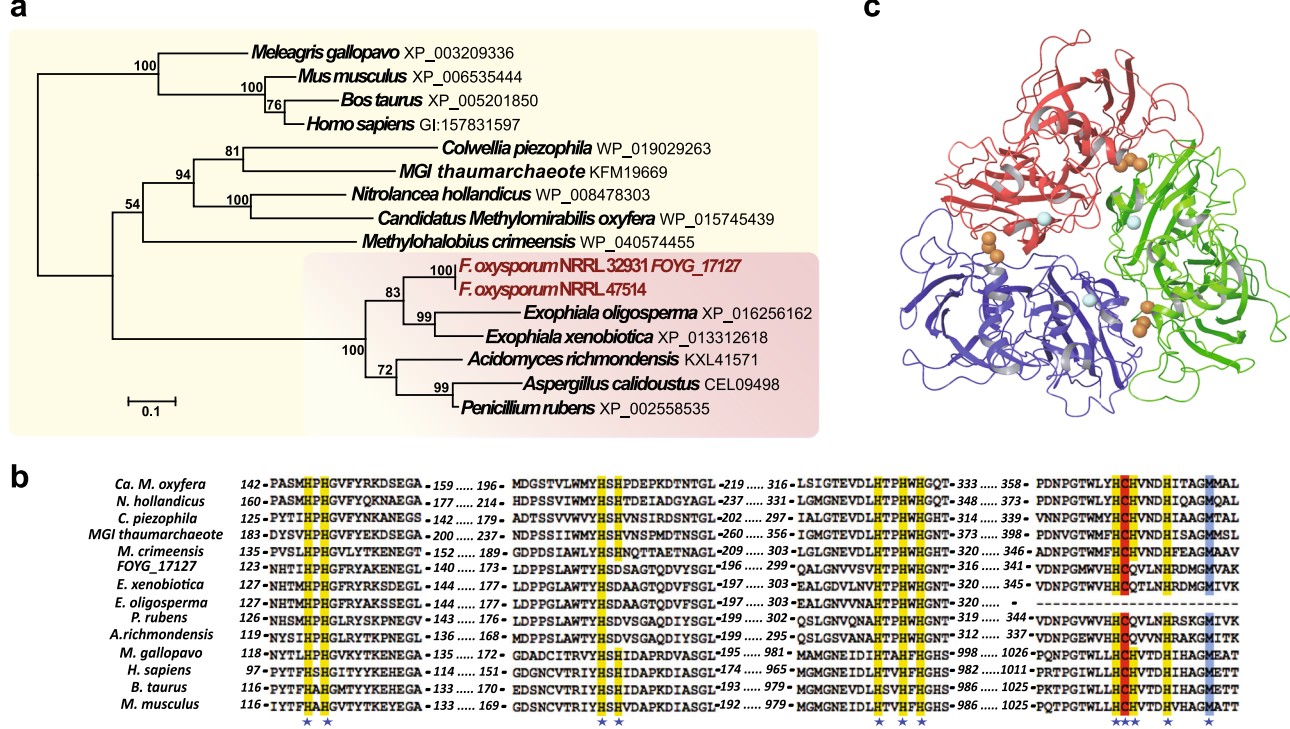

**Fig. 5 Evidence for horizontal transfer of a ceruloplasmin homolog (FOYG_17127) to the LS region of NRRL 32931. (a)** Phylogeny of FOYG_17127 and its homologs, including seven ascomycete fungal homologs (pink), five genes from bacteria and archaea from extreme environments, and concatenated sequences of multicopper oxidase domain-1 and domain-6 of human ceruloplasmin and the corresponding sequences in cow, mouse, and turkey genomes. MGI: Marine group I. Bootstrap values are indicated above the tree branches. **b** Amino acid sequence comparison of four homologous regions of the multicopper blue proteins among FOYG_17127 homologs used for phylogenetic analysis. Numbers proximal to each sequence represent the positions of the amino acid residues of the proteins. *Conserved amino acids specifically involved in copper binding based on the protein structure of human ceruloplasmin[49,55]. **c** Ribbon diagram of a trimeric model of the *F. oxysporum* homolog FOYG_17127 that was created on the basis of human ceruloplasmin crystal structure 1KCW[56], a laccase from *Streptomyces* collector (3CG8)[57], and a blue copper oxidase from *Nitrosomonas europaea* (3G5W) ENREF_87[101]. Residues binding to trinuclear clusters (brown) are in unit A (H127, H129, Y133, H183, and D185) and unit B (H311, H313, H350, and Q352). Residues contributing to copper (blue) binding are in unit A (H308, C351, H356, and M361). An additional binding to an exposed ion (light blue) could be bound by residues in unit A: N123, H124, S186, D191, and Y240 based on the type of nomenclature used[101].

laccases, which were reported as virulence factors in another human pathogenic fungus *C. neoformans*[62].

Collectively, the discontinuous distribution of this group of copper-binding proteins and the presence of highly conserved active binding sites suggest the dynamic nature and functional importance of this group of multicopper oxidases in unique environmental conditions.

**Expansion of other protein families**. LS chromosomes were also shown to contribute to the expansion of protein kinases among FOSC genomes[63]. Among plant pathogenic *F. oxysporum* genomes, expansions were observed within the histidine kinase family, which senses environmental signals, and the TOR kinase, which mediates cell growth[63]. In terms of kinases, the human pathogenic strain NRRL 32931 exhibits a distinct signature and encodes a single TOR kinase—the lowest number of histidine kinases among all the FOSC genomes examined. However, it contains the highest number of HAL kinases (7) and serine/arginine protein kinase-like (SRPKL) kinases (13), which regulate primary potassium pumps[64,65] and mRNA splicing[66,67], respectively. SRPKL kinases are also expanded in dermatophyte fungi[68] and *Coccidioides immitis* (19 copies), another human pathogen. Comparative kinome analysis suggests that LS sequences underwent convergent evolution, resulting in an enhanced and unique capacity for environmental perception and associated downstream responses. The expansion of different kinase families in NRRL 32931 and phytopathogenic *F. oxysporum* strains may well reflect the distinct environment of the human body compared with the plant host.

**Duplication of the ergosterol biosynthesis pathway**. Clinical *Fusarium* species exhibit universal resistance to most antifungals, particularly azoles[69,70]. Several antifungal drugs, such as the azoles, target the sterol biosynthesis pathway, which produces ergosterol, a major constituent of the fungal plasma membrane[71,72]. Interestingly, the entire sterol biosynthesis pathway is extensively duplicated in different *Fusarium* species (Table 4 and Supplementary Data 10). The major azole target, lanosterol 14α-demethylase (*ERG11*), was present in three or more copies in the *Fusarium* genomes. A correlation between *ERG11* amplification and the acquisition of azole resistance in a gene copy number-dependent manner was previously reported for *C. albicans*[73,74]. The contribution of the observed sterol biosynthesis pathway duplication to *Fusarium* resistance, to different azole drugs awaits experimental validation.

The genus *Fusarium* includes many agriculturally important plant pathogens. In addition to vascular wilts caused by *F. oxysporum*, *Fusarium* head blight caused by *F. graminearum* is a major limiting factor of global wheat (*Triticum aestivum*) production, whereas kernel and ear rot of maize (*Zea mays*) caused by *F. verticillioides* occur in almost all regions where maize is grown. Collectively, our comparative study of several sequenced *Fusarium* genomes suggests that species within this genus, which diverged from its sister genus *Cylindrocarpon* ~90 million years ago[75], have evolved natural resistance to antifungals either by amplifying their targets (i.e., those targeting the ergosterol biosynthesis pathway) or by reducing their accumulation within the cell.

Due to limited availability of antifungals, the same classes of fungicides (mostly azoles) used to treat patients with clinical infections are also widely deployed by farmers to control plant diseases. Consequently, the agricultural use of azoles has become a driving force of the antifungal resistance observed in the clinic and was blamed for medical treatment failure, especially among azole-naive patients[6,76,77].

Active efflux of drugs, another mechanism of antifungal resistance, is mainly accomplished by the ABC (ATP-binding

**Table 4 Duplication number of ergosterol biosynthesis pathway genes.**

| Gene ID | 32931 | 34936 (Fol4287) | 26406 | 54003 | 54002 (Fo47) | 26381 | 66176 | 54008 | 54005 | 37622 | 25433 | 54006 | FV | FG | FS | NC |
|---|---|---|---|---|---|---|---|---|---|---|---|---|---|---|---|---|
| ERG11 | 3 | 3 | 3 | 3 | 3 | 4 | 3 | 3 | 3 | 3 | 3 | 3 | 3 | 3 | 3 | 1 |
| ERG24 | 3 | 4 | 4 | 4 | 3 | 4 | 3 | 3 | 3 | 3 | 3 | 3 | 3 | 3 | 3 | 1 |
| ERG26 | 3 | 3 | 3 | 3 | 3 | 3 | 3 | 3 | 3 | 3 | 3 | 3 | 3 | 2 | 2 | 1 |
| ERG6 | 2 | 2 | 2 | 2 | 3 | 3 | 2 | 2 | 2 | 2 | 2 | 2 | 2 | 2 | 2 | 2 |
| ERG3 | 2 | 2 | 2 | 2 | 4 | 2 | 2 | 2 | 2 | 2 | 3 | 2 | 2 | 2 | 2 | 1 |
| ERG5 | 2 | 2 | 2 | 2 | 2 | 2 | 1 | 1 | 1 | 1 | 2 | 1 | 1 | 1 | 1 | 1 |
| ERG4 | 2 | 2 | 1 | 1 | 2 | 2 | 1 | 1 | 1 | 1 | 1 | 1 | 1 | 1 | 2 | 1 |
| ERG25 | 1 | 1 | 1 | 1 | 3 | 2 | 1 | 1 | 1 | 1 | 1 | 1 | 1 | 1 | 1 | 1 |
| ERG2 | 1 | 1 | 1 | 1 | 1 | 1 | 1 | 1 | 1 | 1 | 1 | 1 | 1 | 1 | 2 | 1 |
| ERG1 | 1 | 1 | 1 | 1 | 1 | 1 | 1 | 1 | 1 | 1 | 1 | 1 | 1 | 1 | 1 | 1 |
| ERG7 | 1 | 1 | 0 | 0 | 0 | 1 | 0 | 0 | 0 | 0 | 0 | 0 | 1 | 0 | 2 | 1 |
| ERG27 | 0 | 0 | 0 | 0 | 0 | 0 | 0 | 0 | 0 | 0 | 0 | 0 | 0 | 0 | 0 | 1 |

The five-digit strain number represents the accession number of *F. oxysporum* strains used in Fig. 1. FV *Fusarium verticillioides*; FG *Fusarium graminearum*; FS *Fusarium solani*; NC *Neurospora crassa*.

cassette) or major facilitator superfamily transporter super-families. Similar to other *Fusarium* genomes[24,78], the NRRL 32931 genome encodes 70 ABC transporters—substantially more than other fungal species. Among these, the PDR/ABCG family, which contributes to resistance to xenobiotic compounds, has the highest number of representatives in the NRRL 32931 genome (27), followed by MDR/ABCB (20) and MRP/ABCC (18).

## Discussion

An opportunistic fungal infection is associated with a patient with impaired immunity. It is unclear whether pathogen adaptation is important for such infections to occur. The functional affiliation of the LS-localized genes in strain NRRL 32931 and NRRL 47514 set those human pathogens apart from plant pathogenic strains, supporting the notion that the human-infecting *F. oxysporum* isolate either evolved or acquired virulence-associated genes to establish infection in the hostile environment of the mammalian host. As reported for phytopathogenic *F. oxysporum* genomes[23,24], niche adaptation in human pathogenic *F. oxysporum* genomes appears to have been accomplished in part through the acquisition of transposon-rich LS chromosomes (Fig. 2). This study reports for the first time, to our knowledge, an association between fungal LS chromosomes and potential fungal pathogenicity in human hosts.

These transposon-rich LS chromosomes offer distinct structural and functional compartmentalization within a genome and offer hotspots for recombination and frequent genetic exchanges, serving as a mechanism for rapid gain- or loss-of-infection-related genes, which in turn could accelerate pathogen evolution.

Even though fungal infections caused by *Fusarium* spp. are associated with high mortality rates[6,7], they are mostly limited to immunocompromised patients. In plant pathogenic *Fusarium* strains, horizontally transferred LS chromosomes encode host-specific virulence factors, such as secreted effectors, which effectively suppress plant innate immunity and facilitate plant disease. By contrast, most human-infecting *F. oxysporum* isolates, including NRRL 32931, do not appear to be adapted to overcome host immunity. The patient from whom strain NRRL 32931 was isolated and who is now in continuous complete remission, was able to clear the fungal infection from her bloodstream due to recovery of her immune system and timely disease management, including source control.

However, a recent study suggests that *F. oxysporum* could survive in organs of immunocompromised and immunocompetent mice in the form of thick-walled chlamydospores[79]. Therefore, it is a dangerous possibility that the fungus may develop resistance to the innate immune system over time. As members of the FOSC exhibit pleiotropic resistance to most antifungals, the prospect of some strains developing an increased capacity to overcome host immunity calls for intensified research into the molecular mechanisms of species divergence and adaptation among this group of pathogens. In the compartmentalized *F. oxysporum* genome, genes that contribute to host adaptation are present in the LS chromosomes, providing focal points for studies of pathogenicity. Deciphering the genetic mechanisms that underpin fusarioses will contribute to our efforts to control opportunistic fungal infections.

## Methods

**Fungal isolates**. The *F. oxysporum* human strain isolated from blood is available upon request from the ARS Culture Collection, Peoria, IL (NRRL 32931); the University of Texas Health Science Center at San Antonio (UTHSC 99–853); and the Fungal Genetics Stock Center, Kansas City, MO (FGSC 10444). NRRL 47514 (MRL 8996) was isolated from a patient with contact lens-associated fungal keratitis at Cleveland Clinic Foundation[28].

**Optical mapping**. NRRL 32931 protoplasts were washed three times using phosphare-buffered saline buffer to remove the storage buffer and glycerol, and then lysed in Tris-EDTA buffer (pH 8.0) with 5 mM EGTA and 1 mg/ml proteinase K by heating the protoplast suspension to 65 °C for 30 min. The DNA solution was then incubated at 37 °C overnight, to ensure full digestion of proteins from the lysed protoplasts and the autodigestion of excess proteinase K. Lambda DNA (final concentration ~30 pg/μl) was added to the genomic DNA solution as a sizing standard. DNA solutions were loaded into a silastic microchannel device[80,81] and the DNA molecules were stretched and mounted onto mapping surfaces through capillary action. Mounted DNA molecules were digested with restriction endonuclease BsiWI in NEB buffer 2 (50 mM NaCl, 10 mM Tris-HCl, 10 mM MgCl₂, 1 mM dithiothreitol, pH 7.9; New England Biolabs) with 0.02% Triton X-100, but without bovine serum albumin. Digested DNA molecules were then stained with 12 μL of 0.2 μM YOYO-1 solution (5% YOYO-1; Molecular Probes, Eugene, OR; in TE containing 20% β-mercaptoethanol). Fully automated imaging workstations[80–83] were used to generate single molecule datasets (Rmaps). An optical map spanning the entire genome was constructed using the map assembler[84,85] employing divide-and-conquer and iterative assembly strategies for distributing the computational load[81,83,86]. The assembled optical map coverage for each chromosomal optical map contig and the chromosomal optical map contig sizes are listed in Table 2.

**Pulsed-field gel electrophoresis**. Pulsed-field gel electrophoresis was performed as described previously[24]. Briefly, plugs containing $4 \times 10^8$ protoplasts/ml were loaded on a gel apparatus (1% Bio-Rad Pulsed Field Certified Agarose (FMC, Philadelphia, PA, USA) in 0.5 × TBE) and run using switch times between 60 s and 120 s at 6 V/cm, at a 120° angle for 24 h. Chromosomes of the *Saccharomyces cerevisiae* marker strain (Sc STD) were used as molecular size markers (Bio-Rad, Philadelphia, PA, USA).

**Genome sequencing and assembly**. The NRRL 32931 genome was sequenced using a whole-genome shotgun approach with Illumina technology. A total of 34,374,476 sequence reads were generated, providing over 180× sequence coverage. The NRRL 32931 assembly was generated using ALLPATHS-LG versions 36504 with default parameters[30] and is available at NCBI (AFML01000000). The default k-mer (K) size in ALLPATHS-LG was 96. The assembly was screened against an NCBI mitochondrial database to identify and remove mitochondrial contigs. The genome size was estimated by establishing the frequency of occurrence of each 17 bp k-mer (a unique sequence of 96 (k-mer) nucleotides in length) using a modification of the Lander–Waterman algorithm in ALLPATHS-LG, where the haploid genome length in base pairs was $G = (N*(L - K + 1) - B)/D$, where $N$ is the read length sequenced in base pairs, $L$ is the mean length of sequence reads, $K$ is the k-mer length (17 bp), $B$ is the number of k-mers occurring fewer than four times, and $D$ is the peak value of k-mer.

Genomic DNA of NRRL 47514 was extracted and sequenced using an Illumina NextSeq 500 platform at the University of Massachusetts Amherst Genomics Resource Laboratory and the PacBio RS II platform at the Yale University Genomics facility. The sequencing quality was assessed via FastQC v0.11.5 and the genome was assembled using both sequencing data via SPAdes v3.9.1. The initial assembly was improved using Quiver (in smrtanalysis v2.2.0) and the custom code described in Ayhan et al.[87]. The final assembly was manually inspected for any scaffolding errors using aligned reads and contigs smaller than 1 kb were removed. The raw reads and the genome were deposited in the NCBI database (accession number PRJNA554890). Finally, the completeness of all the assemblies was confirmed by a BUSCO test using the released fungal database (odb9 version)[88].

**Gene structural and functional annotation**. We used a large collection of RNA-Seq/EST data, including 15 strand-specific and paired read datasets generated at the Broad Institute (source organisms: *F. oxysporum*), 18 non-stranded and paired RNA-Seq datasets (source organisms: *F. oxysporum* f. sp. *lycopersici* Fol4287 (NRRL 34936), *F. solani* f. sp. *pisi* NRRL 45880, and *F. oxysporum* f. sp. *pisi* NRRL 37622) from collaborators, plus 4 EST datasets (source organism: *F. oxysporum* f. sp. *cubense* II5 = NRRL 54006). Four Illumina HiSeq runs were generated at the Broad Institute and can be accessed at NCBI under SRX025824, SRX025823, SRX026545, and SRX027736. We used a Trinity transcriptome assembler to process the individual RNA-Seq dataset, to generate transcript assemblies, and then combined all the strand-specific assemblies into one FASTA sequence file and all the non-stranded assemblies and the ESTs into another FASTA sequence file. We then used PASA[89] to align the two transcript files to the genome assemblies, to generate PASA alignments, one for the strand-specific dataset and the other for the non-stranded dataset and the ESTs. Combining individual Trinity assemblies for the PASA alignment resolved a major memory problem caused by combining all of the raw read BAM files into a single BAM file before assembling them using the Trinity assembler.

For gene prediction with EVM[90], we generated ab initio gene models using predictions from GeneMarkES[91], GeneId[92], Augustus[93], GlimmerHMM[94], and SNAP[95], in conjunction with the strand-specific PASA[89] alignment and GeneWise[96] features from a BLAST query of the UniRef90 database. The EVM gene models were first updated with PASA alignments from the 39-stranded RNA-Seq dataset and the output was updated again with PASA alignments from the

22-non-stranded RNA-Seq/EST dataset. The resulting track was filtered to remove spurious genes from the repeat sequences (based on TransposonPSI prediction, repeat PFAM domains, BLAST hits to RepBase, and CDS alignment to >10 different locations of the genome). Additional gene models were added from non-overlapping open reading frames (ORFs) from the 39-stranded RNA-Seq dataset and from the 22-non-stranded RNA-Seq/EST dataset. We also generated a track of EVMLITE gene models from PASA ORFs and the ab initio gene models, and used these to add back additional genes if a gene model did not overlap with the EVM gene models, but was present in OrthoMCL clusters with at least two genomes. For genomes that have previously been annotated (FO2, FG3, and FV3), the old gene models were repeat-filtered and included in the final gene set if a gene model did not overlap with existing gene models. Gene Ontology terms for genes in the whole genome and LS regions were assigned by searching AgBase and Fisher's exact test was performed to identify enriched GO terms ($p$-value < 0.05 and Benjamini and Hochberg false discovery rate < 0.05).

**Phylogenetic analysis**. To determine the phylogenetic placement of clinical isolates NRRL 32931 and NRRL 47514, nucleotide sequences of 55 conserved single-copy orthologous genes (Supplementary Data 1) were selected based on genes recommended by the Fungal Tree of Life and individually aligned using ClustalW. After manual removal of regions with poor sequence quality in any strain, the alignments were concatenated into a single supermatrix (Supplementary Data 2). The general time reversible model was selected for maximum likelihood analyses in MEGA (v7.0.20) with bootstrap test of 500 replicates. ClustalW was also used to align sequences for evolution analysis of other molecules in this study. The aligned sequences were used as input in MEGA (v7.0.20) to generate a maximum likelihood phylogenetic tree using the general time reversible model of nucleotide evolution and JTT (Jones-Taylor-Thornton) model of amino acid evolution with bootstrap test of 500 replicates.

**Eliminating the core genome**. The whole NRRL 32931 genome was used to conduct a BLAST query of the *F. oxysporum* f. sp. *lycopersici* reference genome. A 2 kb sliding-window identity analysis was conducted across the whole genome using the BLAST result. According to the inflection of distribution of all window identities, a 92% window identity was employed as the cutoff for the core genome separation against the core of the *F. oxysporum* reference genome Fol4287. Supercontigs in which more than half of the sequences shared over 92% nt identity with the reference were defined as core regions. The remaining supercontigs were counted as LS regions. NRRL 47514 contigs were aligned to the reference genome Fol4287[87] via MUMmer v3.22 and divided into core or LS sequences according to the alignments.

**Sequencing and analysis of NRRL 32931 mRNA**. Trinity was used to assemble 75 bp paired-end sequencing reads generated on the Illumina sequencing platform. Reads were trimmed for bases with a quality score of <30 and a minimum length of 35 bases using Trimmomatic[97]. Trimmed reads were assembled de novo using Trinity. Transcripts were then mapped to the assembled reference and gene annotations using BLAST, with a minimum *e*-value of 1E-20. A BLAST search was conducted with gene transcripts that failed to align with the assembly or annotations against the NCBI non-redundant (nr) database, the protein database, or known repeat sequences with a minimum $E$-value of $1E-20$ to identify the number of missing *Fusarium* gene sequences. The remaining sequences were run through the NCBI's ORF finder or compared with known repeat sequences to filter out potential transposable elements. Trinity-assembled transcripts that were not annotated as protein coding genes were used to query the Rfam database (http://rfam.xfam.org/search) using the cmscan program[98].

**Repetitive sequences and transposable elements**. Repetitive sequences in the NRRL 32931 and NRRL 47514 genome were identified with the RepeatScout[99] program using default parameters. The repeat families from RepeatScout served as the library for RepeatMasker[100] to determine the frequency and location of each repeat family in the assembly. The repeat families that had more than 20 copies in the assembled genome or had a homolog of a known transposable element in a public database were kept as repetitive sequences for downstream transposon analysis. Transposons were classified by BLASTX against the nr database and structure analysis was performed using REPCLASS with the repeat family sequences.

**Pairwise genome comparisons**. Pairwise genome comparisons between Fol4287, NRRL 32931, and NRRL 47514 were performed using MUMmer v3.22 with a minimum alignment length of 500 bp and otherwise default settings. Alignments of NRRL 32931 supercontig_1.20 and NRRL 47514 LS contigs were performed using Mauve v20150226.

**Alkaline experiment assay**. The wild-type *F. oxysporum* NRRL 32931 isolate was used in the experiments. Microconidia ($1 \times 10^8$) were germinated for 12 h in 2× yeast extract peptone dextrose media. The pH 4 and pH 6 buffered conditions were achieved by adding citrate-$Na_2HPO_4$, whereas the pH 8 buffered condition was achieved by adding potassium phosphate buffer. Mycelia and microconidia were

collected by filtration after 8 h and homogenized using a Bullet Blender (Next Advance, Inc., Averill Park, NY, USA). RNA was extracted using Trizol and cDNA was synthesized using an iScript gDNA Clear cDNA Synthesis Kit (Bio-Rad, Hercules, CA, USA). qRT-PCR was performed using a PerfeCTa SYBR Green SuperMix Low ROX Kit (Quantabio, Beverly, MA, USA). *GAPDH* was used as an internal control gene. Three biological replicates were conducted.

**PacC-eGFP constructs**. *pacC* genes were amplified from 1000 bp upstream of the starting codon to before the predicted cleavage site and stop codon in *pacC_O* and *pacC_b*, respectively. The products were inserted into the pENTR entry vector using the pENTR™/D-TOPO™ Cloning Kit (Invitrogen, K240020, Carlsbad, CA, USA). The LR reaction was performed using the pENTR construct with the pFPL-Gh destination vector (Addgene Plasmid #61648, Cambridge, MA, USA). The correct plasmids were selected via Sanger sequencing and transformed into AGL1 *Agrobacterium*-competent cells. *Agrobacterium*-mediated transformation was conducted as described. Individual fungal transformants were collected via single spore isolation and genomic DNA was extracted for PCR verification (FOYG_17127 F: 5′-CACTTAATCAACTCTTCATACC-3′, R: 5′-TCAACTAGCC TGATACTTC-3′; *pacC_O* F: 5′- CACCGGTCTTGGCAGTCTGGGGCCAA-3′, R: 5′-ACCGCCGGCAGCAGCGCTGTATCC-3′; *pacC_b*: F: 5′-CACCGGCGAAG GCTAAGGACAGCAA-3′, R: 5′-CTGTCTCAGTGAGTTCAGGCGTG-3′).

**Homology modeling of FOYG_17127**. Homology modeling studies were conducted based on high-resolution structures of proteins determined to be the most closely related to the fungal protein by NCBI BLAST: human CP (PBD ID: 1KCW with 3.00 Å resolution and 37% identity, 54% similarity, 8% gaps)[56] and a laccase from *Streptomyces* collector (PBD ID: 3CG8 with 2.68 Å resolution and 23% identity, 34% homology, 28% gaps)[57]. On the basis of sequence alignment between the fungal protein and the two C-terminal domains of human CP, Prime Build Homology Model produced a preliminary secondary structure model complete with the copper cofactors retained by the conserved sites. The quaternary structure of FOYG_17127 was produced by the backbone alignment of three copies of the resulting model onto the homotrimeric template of the laccase from *Streptomyces* collector. Global minimization of the resulting assembly, using the OPLS_2005 force field implemented within Schrödinger software Prime, furnished the final homology model.

**Statistics and reproducibility**. Sample sizes were determined based on previous published studies and pre-experiments. All phylogenies were tested by boot-strapping with 500 replicates. Fisher's exact test was used for testing enriched GO terms. *P*-value < 0.05 was used as cutoff to indicate statistical significance. For qRT-PCR experiment to quantify *pacC* gene expression, three biological replicates were performed. Two-sided Student's *t*-test was used for statistical analysis of expression results.

**Reporting summary**. Further information on research design is available in the Nature Research Reporting Summary linked to this article.

## Data availability
The raw sequencing data used for de novo whole-genome assembly of NRRL 32931 are available at NCBI (SRX101569, SRX101560, SRX101558, SRX081496, SRX081494, and SRX081474). The accession numbers of NRRL 47514 (MRL 8996) whole-genome sequencing data are SRX6453258 (Illumina) and SRX6453257 (PacBio RS II). All Illumina HiSeq sequences of RNA-Seq are available with accession numbers SRX026545, SRX027736, SRX025824, and SRX025823 at NCBI. All other data are available upon request.

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

## Acknowledgements

We thank Dr Bruce Birren and the Broad Institute manual and automated annotation teams for their support on this and many other fungal genomic projects; Dr Frederick M. Ausubel for critical review of the manuscript; and Cindy Zhang from Boston University for drawing the diagrams shown in Fig. 1. The mention of firm names or trade products does not imply that they are endorsed or recommended by the US Department of Agriculture over other firms or similar products not mentioned. The USDA is an equal opportunity provider and employer. This project was supported by the National Research Initiative Competitive Grants Program Grant numbers 2008-35604-18800 and MASR-2009-04374 from the USDA National Institute of Food and Agriculture, the National Eye Institute of the National Institutes of Health (R01EY030150) and grant BIO2016-78923-R from the Spanish Ministerio de Economía y Competitividad. Data were analyzed at the Massachusetts Green High Performance Computing Center (MGHPCC). L.-J.M. is also supported by an Investigator Award in Infectious Diseases and Pathogenesis by the Burroughs Wellcome Fund BWF-1014893.

## Author contributions

L.-J.M., H.C.K., M.R., and A.D.P. designed this study. Y.Z. and L.-J.M. wrote the manuscript with contributions from all authors. I.Y., A.M., N.W., and E.P. provided strains. Y.Z., D.H.A., L.G., G.A.D., and S.S. performed analysis. H.Y., D.T., and K.B. provided functional and experimental validations. S.G.Z. and D.S. performed optical mapping. K.O. contributed FOSC phylogeny. D.H.A., J.J., T.S., S.Y., and Q.Z. provided genome sequencing and assembly.

## Competing interests

The authors declare no competing interests.
