## [Peer Review File · Communications Biology]

Reviewers' comments:

Reviewer #1 (Remarks to the Author):

In this manuscript, Zhang and collaborators report on phenotypic and genomic differences between isolates of *Fusarium oxysporum* infecting humans, plants and a non-pathogenic isolate.

I have no comments on comparative genomic analyses or fitness assays. The findings reported represent a huge amount of work and in my opinion analyses and experiments have been competently carried out.

My main criticisms are centred on the focus on so-called lineage specific chromosomes, and on sample size and sets of samples used in comparison.

Chromosomes specific to NRRL 32931 are qualified as « Lineage specific », but the authors actually provide no evidence that they are specific to a lineage. L94 actually suggest that there is actually no such thing as « lineages » of human pathogenic *Fusarium oxysporum*. More generally it's not clear whether the authors' conclusions about differences in fitness or genomic differences are valid for NRRL 32931 only, for a yet-to-be characterized lineage to which NRRL 32931 would belong, or to all clinical isolates of *Fusarium oxysporum*.

What is the set of genomes used in comparisons is not clear throughout the manuscript. Is it the genomes of isolates shown in Figure 1? So why do comparisons only mention phytopathogenic species (e.g. L210, L320)? no comparison to the non-pathogenic isolate 54002?

Minor comments:

L100: Authors show that clinical isolate displays certain traits that are *potentially* adaptative, but they do not show that these traits are adaptive. No adaptation experiment was conducted. Adaptation is an hypothesis.

Reviewer #2 (Remarks to the Author):

This paper entitled "The genome of an opportunistic fungal pathogen *Fusarium oxysporum* carries a unique set of lineage-specific chromosomes" describes the discovery of four extra chromosomes in a strain of *F. oxysporum* isolated from the human bloodstream, and absent from other *F. oxysporum* genomes (all of them? I am not really sure, this is not very clear). These chromosomes are referred as "lineage-specific" or LS chromosomes. Some LS chromosomes were previously found in phytopathogenic *F. oxysporum* genomes, and were shown to have been acquired through horizontal transfers and to play a role in host-specific pathogenicity. Here, the authors performed optical mapping and sequenced the whole genome and the transcriptome of this *F. oxysporum* strain isolated from an immunocompromised patient. They described the gene content of LS chromosomes in this clinical strain and found gene family expansions (e.g. metal-ion transporters) as well as expressed genes (e.g. a homolog of ceruloplasmin) important for virulence in mammals. The authors also tested experimentally if this clinical strain had fitness advantage at body temperature (37°C) compared to strains isolated from other environments (plants), which informs about host adaptation (to mammals). The authors thus used complementary approaches, i.e. genomics/transcriptomics and laboratory experiments to study the adaptation of a *F. oxysporum* strain to the human host. The findings presented here are interesting. However, I have several concerns .

First, the authors claimed the LS chromosomes are specific to that strain. But did they look for these chromosomes and/or genes contained in these chromosomes in databases? Other genomes of *Fusarium*? It is said in the legend of the Figure 3 that the LS regions lack homologous sequences in the reference genome, nothing else. This information is crucial.

Second, I was expecting to read something about the acquisition of these LS chromosomes in this clinical strain, i.e. horizontally transferred as in other phytopathogenic strains? The authors only showed the evidence of the horizontal acquisition of a single gene (Fig. 6) but they don't discussed about this in the main text (L280-282 but it is unclear). If these extra chromosomes contained genes absent from other *Fusarium* species (or at least *F. oxysporum* strains), a systematic search of their

origin (a search against all organisms sequenced so far) should be done, as the authors did for the ceruloplasmin gene.

Third, there are some methodological issues:

- The table 2 displays the percentage of mapped scaffold on the optical map. I was very surprised to see the very low percentage of mapping on the extra chromosomes. How do the authors explain this? How is it possible?

- The method section is incomplete: there is no information about how the phylogenetic tree in Fig. 1 was constructed (in fact it is in the legend of the Figure 1, where it is not appropriate). Actually, there is no information of how phylogenetic trees (because there are several) have been made in the manuscript.

- The authors tested the resistance of this clinical strain to antifungals, and a comparison with *A. fumigatus* was made. How is it possible to conclude anything by comparing two strains belonging to two different species?

Last, the manuscript could be improved by working on the main text, for removing some parts dispensable in the results but needed in the methods section. For example, from line 164 to 176, it is a description of the method used to assembly and to map the genome; this could be shortened. There are also some mistakes in tense (sometimes present, sometimes past), and some titles are not well chosen, rendering the reading of the manuscript difficult. In particular, the first part entitled "F. oxysporum isolated from blood of a fusariosis patient, displays higher fitness at high temperature under submerged growth conditions". We are thus expecting to find results and discussion about laboratory experiments here. Instead, we have a lot of methodological details (how the strain was isolated, the name of the collections where the strain is available, the phylogenetic relationship of the strain to others). Also, L139, it is written "all isolates were able to germinate and grow well at 37°C in submerged culture", is the title wrong then? In fact, it should be more explained in the paragraph that the clinical strain grew significantly better than others at 37°C, it not clear as it is written. Also, in this title, would it be more appropriate to add "and" between "high temperature" and "under submerged growth conditions"?

In general, I found that results are not sufficiently discussed.

General comments:

- With molecular data it is more appropriate to perform ML analyses than MP. With ML, the user has to select the best-fit model of DNA evolution (modeltest, Posada and Crandall 1998); the information of which model has been chosen, based on what criteria, is missing. Also, there is no information about how phylogenetic trees have been performed in the manuscript, root, bootstrap, method etc.

- In the phylogenetic tree, I can see that several strains have been isolated from tomatoes: do all these strains carry some LS chromosomes? If yes, are they identical/similar? What about the watermelon strain? It would be interesting to discuss this point.

- L94-95: there are several clinical strains which are phylogenetically not related, are the genomes available for some of these strains? Have the authors checked the presence/absence of the LS chromosomes found in the strain NRRL32931 in those other strains?

- Paragraph from L127 to 132: This paragraph is unclear. We expect a discussion on the phylogenetic tree presented in Figure 1. I don't understand why the authors discussed about FOSC 3-A which is not in the tree? And why do they talk about haplotypes? I guess these haplotypes have been defined based on MLST or microsatellites? And in the tree there is no reference to these haplotypes?

Other comments:

L48: virulence in mammals instead of on

L48: "this study provided the first evidence for the genome compartmentalization in a human pathogenic fungus". *Fusarium oxysporum* is not a human pathogenic fungus but an opportunistic pathogenic fungus of plants and animals, as the authors wrote L50. Thus, compartmentalization has already been described in pathogenic fungi, here is not the first evidence.

L91: a tomato pathogenic isolate without "s".

L98: "we characterized *F. oxysporum*"; "sequenced" would be more appropriate

L105-106: "metal ion transport and inorganic cation transporter". Maybe to be homogenized: "metal ion transporter and inorganic cation transporter"

L116-119: should be in the method, not in the results and discussion.

L120-126: I am not sure whether this paragraph is necessary for the understanding of the study.

L124: a problem of tense for remained. How old is the patient today? The 18 years passed or not yet?

L127: the authors began the sentence with "phylogenetically" but they don't discuss about a

phylogenetic tree, but a haplotype. This is not phylogenetics. Genetically?

L128: "Clonal lineage FOSC 3-a". But FOSC 3-a is the name of a strain? So it cannot be a clonal lineage...

L129 and 132: twice Figure 1, remove one

L133: "a generally high level of fitness". "Generally" does not mean anything... statistically? So we expect a pvalue, a statistical test.

L139: all strains grew well at 37°C so the title of this part (the clinal strain displays a higher fitness at thigh temperature under submerged growth conditions) is false. I reckon it is true that the clinical strain grew better than others at 37°C, but this is not what it is written.

L139-141: it is written that 37°C is the condition of the human bloodstream or of plant xylem vessels; I did not know that the plant xylem vessels were at that temperature??

L145-151: why the ability to penetrate a cellophane was not tested at 37°C to mimic bloodstream?

L153-154: Optical mapping is a method to joining scaffolds based on restriction enzymes, not to "investigate whether LS chromosomes are present". This sentence should be modified.

L162/163: this sentence could be removed, redundant with what is written just below

L164-169: should be in the Method, too detailed here.

L170: The supplementary file 3 is "PCR amplification of all PacC alleles in NRRL 32931". This has nothing to do here.

L177: most sequences belonging

L181: "core regions defined as scaffolds". "Scaffold" is a term already existing, and this is not its meaning.

L192: transposons. Which category? Is a category statistically more represented than others?

L211: were

L208: LS genes were distinct.

L223-233: a discussion is missing here, it is just results, I don't get what the authors wanted to highlight here

L238: were

L234-240: again I don't get what the authors wanted to highlight here

L241: lacked

L262: was

L264: a problem with the format of the bibliography

L265-266: do the authors suggest a role of transposable elements in the acquisition of genes in LS chromosomes? There are plenty of examples in the literature and this point could be discussed

L278-282: Horizontal transfer?

L344: Cyllindrocarpon

L347-351: It could be interesting to cite the example of *Candida auris*, a new global emerging threat?

L356: which statistical test? Pvalue?

L362: over is not needed here

L421: 28, 30 and 33

L424: "similar" results. Which statistical test did the authors performed to conclude this?

L543: performed

L545: Sanger

Figure 1: The legend is not a figure legend, and details should be put into the Method section. For example, in a figure legend there is no need to have the name of the software used for aligning sequences or to know the number of steps for the most parsimonious tree. By the way, with molecular data it is more appropriate to perform ML analyses in which we have to select the best-fit model of DNA evolution (modeltest, Posada and Crandall 1998). I don't know which model of DNA evolution was chosen for the ML analysis, it is not written in this legend (where it is not its place) nor in the method section (where it should be, with all details about the phylogenetic analysis). The scale of the tree should be in number of substitutions per site rather than in steps. It could be great to have the number of LS chromosomes in the different strains shown in that tree when information is available, or a figure on the right side representing these LS chromosomes in the different strains.

Figure 2: A) the scale bar is not present in all pictures. It would be more easy for the reader to have the origin of the strain next to the name.

L981 and 983: "before" and "after". I had to read twice to understand the sentences.

Figure 3: L988 no 'of' between genome and Fol4287

Part B: why showing the pulse fielded gel and its inverse image? I think only one is needed they are the same...

L1001: red in lowercase

Figure 5: Part A: maybe the sequences of the clinical strain NRRL 32931 should be in another color to render the figure easy to understand. There is no information about how this phylogenetic tree was made (ML? bootstrap? Root?) All this should appear, and a detailed description should be added in the Method section. The legend does not specify what the right part of the figure 4A represents.

Part C: The sentence is unclear.

Part D: a statistical test to assume that the expression increase with the pH?

Part E: the legend is too short, it does not explain correctly the 4 images.

Figure 6: A (it is written in lowercase on the figure): why did the authors put the name of the gene (which is non-informative except if we would like to retrieve the sequence from the genome) for *Fusarium oxysporum* and the name of the species for all other nodes? There is no pink highlighted box as mentioned in the legend. As in figure 5, how was done the tree? Which method? A detailed description of the analysis should be added in the Method section.

Figure 7: the legend of the color should not be separated in two different graphs. The legend is unclear, what was the experiment? For the Figure 2 the authors provided a lot of information, here nothing. A quick summary of the experiment is necessary for the understanding of the figure.

Reviewer #3 (Remarks to the Author):

In the paper "The genome of an opportunistic fungal pathogen *Fusarium oxysporum* carries a unique set of lineage-specific chromosomes", the authors present their genome analysis of the pathogen *Fusarium oxysporum* from the blood of a patient with leukemia. *Fusarium oxysporum* is known for horizontally acquiring lineage-specific (LS) chromosomes. The authors of this manuscript also focus on the genes found in the LS chromosome – noting that some genes present appear to be correlated with pathogenicity, and state that LS chromosomes of clinical isolates have not been previously reported. The authors also assess growth rates of this and a small number of other isolates, and MIC for some antifungals for just this one isolate. They also present an RNAseq dataset that they use to gauge the completeness of their genome assembly, and look at the expression of a small group of transcription factor homologs (pacC).

Although the authors have done a reasonable amount of work characterising this isolate, the write-up is of a pretty low quality (see my comments for some examples). Figures are also mostly of a low quality (especially Fig 1, 3 and 4). Many of the analysis appear to be lacking a clear overview and presentation, in addition to pulling out the most interesting findings. The RNAseq experiment and the structural predictions appear to be particularly rushed and lacking key details. I'm concerned there has been no multiple correction for the GO-term enrichment, and a table of the uncorrected values discussed are not even presented in the manuscript – despite being one of the main results presented. MICs should also be taken for a panel of clinical and non-clinical *Fusarium* isolates to provide some overview of rang.

Additional comments:

Page 4 line 79-80 "Fusariosis, an invasive fungal infection caused by *Fusarium* spp., is listed as the second most common opportunistic mold infection after aspergillosis 6,7". The two citations are for papers on Fusariosis specifically, and not systematic reviews on the most common opportunistic mold infections as I believe is implied (i.e. listed). I do not see lists of most common infections in those papers, and it's dubious that Fusariosis, which is often described as rare, is more common than Candidiasis or Cryptococcosis for example. Please check or clarify this statement.

Page 4 line 80: "positive blood cultures were detected among approximately 50% of fusariosis patients". How were the other 50% of fusariosis patients diagnosed?

Page 4 line 82: "Because *Fusarium* spp. are broadly resistant to most clinically available antifungals [9]". The citation is to a paper on fusariosis in a Brazilian hospital that does no characterisation of antifungals at all. The paper doesn't even mention the word antifungal. Find a suitable reference for this statement or omit it.

Page 5 line 98: "To address these questions, we characterized *F. oxysporum*". This is very vague. In what way did you characterise it? Also, this sentence at the start of a paragraph refers to the sentence in the previous paragraph, which are not phrased as questions.

Page 5 line 99: "We show here that this" needs rewording.

Page 5 line 106: Presenting a brief summary of the results at the end of the introduction is common, but for brevity, p-values of GO-term enrichment should probably be saved for the results.

Page 5/6 lines 116-119: "The strain is available upon request ..." is not results. Move to methods.

Page 6 line 127: "Phylogenetically, NRRL32931". The 1st and 2nd paragraph was about the isolate and its origin, which seem more suited to the methods if anything, and this paragraph seems to have skipped the sequencing step before launching into phylogenetics. A very short description of the sequencing and computational methods used (assuming they were) to generate a phylogenetic tree is required. Why has this been excluded?

Page 6 line 133: "Phenotypically, ". Delete.

Page 6 line 138: This paragraph is focused on the growth rates of isolates at different temperatures, and says that data is not shown for isolates at 34C. Why is not shown? Also what is Fo47? You say non-pathogenic, but where has it come from?

Page 7 line 164-165: "The genome assembly of NRRL 32931 was generated... and is available at NCBI (AFML01000000)". On NCBI, AFML01000000 is listed as isolate 3-a, and the name "32931" does not appear. Can you confirm these are the same isolate, and if so explain the discrepancy in the names?

Page 7 line 175: "By contrast, mapping of the assembled sequence to the LS chromosomes from 12 to 15 only resulted in 31.1%, 42.5%, 18.5% and 26.4% mapping rates, respectively" needs rewording for clarity. Also, what is the take home message of this?

Page 11 line 214: "Interestingly, ". Delete.

Page 12 lines 262-268: Needs rewording and restructuring. Also are you discussing GO-terms? Because it is listed as only "functional category" which could be any number of things. Also, do these copper-binding proteins have GO terms relating to iron homeostasis? It might be helpful to list all of the "functional category's" that this gene has assigned to it (rather than just the one) to give an indication of its possible function.

Page 13 line 301-302: "we noticed ... is distantly related". Be more specific.

Page 16 line 361: "An opportunistic infection occurs when the host immunity is impaired.". No. An opportunistic infection does not always occur when the host immunity is impaired. Also, opportunistic infections have a wide range of causes – not only impaired immunity.

Page 20 line 463: Provide citations or NCBI accessions for these RNAseq datasets.

Page 20 line 474: "circumvented a major memory problem". Badly described.

Page 21 line 495: You didn't use multiple correction with your Fisher exact test?

Figure 1. Is this ML or MP? Can you explain "250 steps" as a scale bar?

Response to reviewers (COMMSBIO-19-0349-T)

Reviewer #1:

In this manuscript, Zhang and collaborators report on phenotypic and genomic differences between isolates of *Fusarium oxysporum* infecting humans, plants and a non-pathogenic isolate. I have no comments on comparative genomic analyses or fitness assays. The findings reported represent a huge amount of work and in my opinion analyses and experiments have been competently carried out.

Response: We greatly appreciate this reviewer's recognition.

My main criticisms are centred on the focus on so-called lineage specific chromosomes, and on sample size and sets of samples used in comparison. Chromosomes specific to NRRL 32931 are qualified as "Lineage specific", but the authors actually provide no evidence that they are specific to a lineage. L94 actually suggest that there is actually no such thing as "lineages" of human pathogenic *Fusarium oxysporum*. More generally it's not clear whether the authors' conclusions about differences in fitness or genomic differences are valid for NRRL 32931 only, for a yet-to-be characterized lineage to which NRRL 32931 would belong, or to all clinical isolates of *Fusarium oxysporum*.

Response: To specifically address this comment, this revision included another human pathogenic strain, NRRL 47514 (MRL 8996), isolated from a contact lens associated with the USA 2005/06 fusarium keratitis outbreak. We have generated a whole genome assembly combining PacBio and Illumina sequences. The comparative analysis confirmed that these two human pathogens share almost identical LS sequences, including putative pathogenicity-related genes, such as ceruloplasmin homolog and the expansion of PacC genes. None of these genes was observed in any plant pathogenic strain. We have included these data in Figure 4, Table 3, and Supplementary Figure 3.

What is the set of genomes used in comparisons is not clear throughout the manuscript. Is it the genomes of isolates shown in Figure 1? So why do comparisons only mention phytopathogenic species (e.g. L210, L320)? no comparison to the non-pathogenic isolate 54002?

Response: We compared the NRRL 32931 genome with the reference genome *F. oxysporum* 4287 (NRRL 34936), which was labelled as 34936 *lycopersici* in Figure 1. The non-pathogenic strain also has a single LS chromosome (4.7 Mb), confirmed by optical mapping. However, identifying the genes encoded on this chromosome is a different project, which we hope to pursue in the near future.

Minor comments:

L100: Authors show that clinical isolate displays certain traits that are *potentially* adaptive, but they do not show that these traits are adaptive. No adaptation experiment was conducted. Adaptation is a hypothesis.

Response: We agree that adaptation is a hypothesis and we have adjusted our language carefully throughout the manuscript to reflect this.

Reviewer #2:

This paper entitled “The genome of an opportunistic fungal pathogen *Fusarium oxysporum* carries a unique set of lineage-specific chromosomes” describes the discovery of four extra chromosomes in a strain of *F. oxysporum* isolated from the human bloodstream, and absent from other *F. oxysporum* genomes (all of them? I am not really sure, this is not very clear).

Response: Excellent point!

The four extra chromosomes present in NRRL 32931 are absent in all sequenced phytopathogenic *F. oxysporum* genomes. We have made sure this is clearly stated in this revised manuscript.

These chromosomes are referred as “lineage-specific” or LS chromosomes. Some LS chromosomes were previously found in phytopathogenic *F. oxysporum* genomes, and were shown to have been acquired through horizontal transfers and to play a role in host-specific pathogenicity. Here, the authors performed optical mapping and sequenced the whole genome and the transcriptome of this *F. oxysporum* strain isolated from a immunocompromised patient. They described the gene content of LS chromosomes in this clinical strain and found gene family expansions (e.g. metal-ion transporters) as well as expressed genes (e.g. a homolog of ceruloplasmin) important for virulence in mammals. The authors also tested experimentally if this clinical strain had fitness advantage at body temperature (37°C) compared to strains isolated from other environments (plants), which informs about host adaptation (to mammals). The authors thus used complementary approaches, i.e. genomics/transcriptomics and laboratory experiments to study the adaptation of a *F. oxysporum* strain to the human host. The findings presented here are interesting.

Response: We appreciate this reviewer’s recognition.

However, I have several concerns:

First, the authors claimed the LS chromosomes are specific to that strain. But did they look for these chromosomes and/or genes contained in these chromosomes in databases? Other genomes of *Fusarium*? It is said in the legend of the Figure 3 that the LS regions lack homologous sequences in the reference genome, nothing else. This information is crucial.

Response: Excellent point! As stated in our response to the previous comment, the four extra chromosomes in NRRL 32931 are absent in all sequenced *F. oxysporum* genomes.

For this revision, we also conducted a comprehensive analysis of all "LS genes". 1) Not a surprise many LS genes have matches within the *Fusarium* genus, reflecting gene family expansion, as we observed frequently among LS genes (Ma 2019). Figure 4B captures some uniquely expanded gene families in both the blood and keratitis strains, including PacC genes, Hal Kinases, and ion transporters. 2) Distinctively, these two human pathogenic strains also have genes acquired through horizontal transfer, and these are all present in the 4 LS chromosomes. The best example of such a gene, the ceruloplasmin homolog, is illustrated in Figure 7.

Second, I was expecting to read something about the acquisition of these LS chromosomes in this clinical strain, i.e. horizontally transferred as in other phytopathogenic strains? The authors only showed the evidence of the horizontal acquisition of a single gene (Fig. 6) but they don't discussed about this in the main text (L280-282 but it is unclear). If these extra chromosomes contained genes absent from other *Fusarium* species (or at least *F. oxysporum* strains), a systematic search of their origin (a search against all organisms sequenced so far) should be done, as the authors did for the ceruloplasmin gene.

Response: Thanks for pointing this out! In the revised manuscript, we included a keratitis strain MRL 8996 genome in the genomic comparison. The results are now presented in Figure 4 and Table 3. In summary, LS regions are shared between the blood strain NRRL 32931 and the keratitis strain MRL 8996, including the ceruloplasmin homolog and all four PacC genes, which are not shared by the plant pathogenic isolate FoI4287.

Third, there are some methodological issues:

- The table 2 displays the percentage of mapped scaffold on the optical map. I was very surprised to see the very low percentage of mapping on the extra chromosomes. How do the authors explain this? How is it possible?

Response: Thank you for raising this interesting point! Repetitive sequences are overly abundant in the LS chromosomes. They are highly AT-rich. The AT content of all repeats over 1.5 kb in size ranges from 60% to 90%. In addition, the resolution of the optical map is 50 kb, which means that fragments less than 50 kb could not be put into chromosomes. To provide an explanation for this, we added two sentences.

- The method section is incomplete: there is no information about how the phylogenetic tree in Fig. 1 was constructed (in fact it is in the legend of the Figure 1, where it is not appropriate). Actually, there is no information of how phylogenetic trees (because there are several) have been made in the manuscript.

Response: Thanks for the careful reviews. We have moved and added information about the phylogenetic construction into the Materials & Methods section.

- The authors tested the resistance of this clinical strain to antifungals, and a comparison with *A. fumigatus* was made. How is it possible to conclude anything by comparing two strains belonging to two different species?

Response: *A. fumigatus* is the most common mold infection. Even though this comparison is not comprehensive, it is representative.

Last, the manuscript could be improved by working on the main text, for removing some parts dispensable in the results but needed in the methods section. For example, from line 164 to 176, it is a description of the method used to assembly and to map the genome; this could be shortened. There are also some mistakes in tense (sometimes present, sometimes past), and some titles are not well chosen, rendering the reading of

the manuscript difficult. In particular, the first part entitled “*F. oxysporum* isolated from blood of a fusariosis patient, displays higher fitness at high temperature under submerged growth conditions”. We are thus expecting to find results and discussion about laboratory experiments here. Instead, we have a lot of methodological details (how the strain was isolated, the name of the collections where the strain is available, the phylogenetic relationship of the strain to others). Also, L139, it is written “all isolates were able to germinate and grow well at 37°C in submerged culture”, is the title wrong then? In fact, it should be more explained in the paragraph that the clinical strain grew significantly better than others at 37°C, it not clear as it is written. Also, in this title, would it be more appropriate to add “and” between “high temperature” and “under submerged growth conditions”?

Response: Thank you for the excellent suggestions. We have carefully revised the main text and moved the description of a method in line 164 to 176 in the Materials & Methods section. Also, we have condensed the genome assembly section as the reviewer suggested. Some titles in the manuscript have been changed to make it easier to read. For instance, the title of the first part was changed to “*F. oxysporum* NRRL 32931 displays fitness at high temperature and under submerged growth conditions”.

General comments:

- With molecular data it is more appropriate to perform ML analyses than MP. With ML, the user has to select the best-fit model of DNA evolution (modeltest, Posada and Crandall 1998); the information of which model has been chosen, based on what criteria, is missing. Also, there is no information about how phylogenetic trees have been performed in the manuscript, root, bootstrap, method etc.

Response: We appreciate this suggestion and have adopted ML for all phylogenetic analyses, including those shown in Fig. 1, Fig. 6, and Fig. 7, along with bootstrap values for all. Detailed methods are included in the Materials & Methods section now.

- In the phylogenetic tree, I can see that several strains have been isolated from tomatoes: do all these strains carry some LS chromosomes? If yes, are they identical/similar? What about the watermelon strain? It would be interesting to discuss this point.

Response: Very interesting questions, indeed! For the three tomato strains included in Fig. 1, 54003 has chr14, one of the LS chromosomes that contains the hallmarks of *F. oxysporum* pathogenic effectors (SIX genes) (our unpublished results). In our 2010 paper¹, we have demonstrated that transferring chr14 could convert a non-pathogenic strain to a pathogenic one. The other tomato strain, 26381, a root rot instead of a wilt pathogen, causes disease through a different mode of action. The profile of LS encoding effectors and association with race structure among melon strains was published², demonstrating the presence of LS chromosomes. LS chromosomes are also reported in other *formae speciales*, including legume³, melon², onion⁴, and multiple cucurbit species⁵.

1. Ma LJ, Van Der Does HC, Borkovich KA, et al. Comparative genomics reveals mobile pathogenicity chromosomes in *Fusarium*. *Nature*. 2010;464(7287):367-373. doi:10.1038/nature08850
2. Schmidt SM, Lukasiewicz J, Farrer R, van Dam P, Bertoldo C, Rep M. Comparative genomics of *Fusarium oxysporum* f. sp. *melonis* reveals the secreted protein recognized by the Fom-2 resistance gene in melon. *New Phytol*. 2016. doi:10.1111/nph.13584
3. Williams AH, Sharma M, Thatcher LF, et al. Comparative genomics and prediction of conditionally dispensable sequences in legume-infecting *Fusarium oxysporum* formae speciales facilitates identification of candidate effectors. *BMC Genomics*. 2016;17(1). doi:10.1186/s12864-016-2486-8
4. Taylor A, Vágány V, Jackson AC, Harrison RJ, Rainoni A, Clarkson JP. Identification of pathogenicity-related genes in *Fusarium oxysporum* f. sp. *cepae*. *Mol Plant Pathol*. 2016;17(7). doi:10.1111/mpp.12346
5. Van Dam P, Fokkens L, Ayukawa Y, et al. A mobile pathogenicity chromosome in *Fusarium oxysporum* for infection of multiple cucurbit species. *Sci Rep*. 2017;7(1). doi:10.1038/s41598-017-07995-y

- L94-95: there are several clinical strains which are phylogenetically not related, are the genomes available for some of these strains? Have the authors checked the presence/absence of the LS chromosomes found in the strain NRRL32931 in those other strains?

Response: In this revision, we included another *F. oxysporum* clinical strain, isolated from a contact lens associated with the USA 2005/06 fusarium keratitis outbreak. The genomic comparison revealed that the putative pathogenicity related genes in the blood strain, such as ceruloplasmin homolog, PacC genes, were also present in this fusarium keratitis strain, but absent from all plant pathogenic strains.

- Paragraph from L127 to 132: This paragraph is unclear. We expect a discussion on the phylogenetic tree presented in Figure 1. I don't understand why the authors discussed about FOSC 3-A which is not in the tree? And why do they talk about haplotypes? I guess these haplotypes have been defined based on MLST or microsatellites? And in the tree there is no reference to these haplotypes?

Response: We have revised the manuscript and removed irrelevant parts accordingly.

Other comments:

L48: virulence in mammals instead of on **Response:** Changed. Thanks.

L48: "this study provided the first evidence for the genome compartmentalization in a human pathogenic fungus". *Fusarium oxysporum* is not a human pathogenic fungus but an opportunistic pathogenic fungus of plants and animals, as the authors wrote L50. Thus, compartmentalization has already been described in pathogenic fungi, here is not the first evidence.

Response: The sentence was corrected to "This study provides the first evidence for genome compartmentalization in an opportunistic human pathogenic fungus

and suggests that LS chromosomes play an important role in host adaptation”.

L91: a tomato pathogenic isolate without “s”. **Response: Changed. Thanks.**

L98: “we characterized F. oxysporum”; “sequenced” would be more appropriate
Response: Changed. Thanks.

L105-106: “metal ion transport and inorganic cation transporter”. Maybe to be homogenized: “metal ion transporter and inorganic cation transporter”
Response: Changed. Thanks.

L116-119: should be in the method, not in the results and discussion.
Response: Changed. Thanks.

L120-126: I am not sure whether this paragraph is necessary for the understanding of the study.

Response: This paragraph provides a phylogenetic framework of human pathogenic strains.

L124: a problem of tense for remained. How old is the patient today? The 18 years passed or not yet?

Response: The patient was 3 years old at the onset of the case. Eighteen years later, she is now 21.

L127: the authors began the sentence with “phylogenetically” but they don’t discuss about a phylogenetic tree, but a haplotype. This is not phylogenetics. Genetically?

Response: Changed. Thanks.

L128: “Clonal lineage FOSC 3-a”. But FOSC 3-a is the name of a strain? So it cannot be a clonal lineage...

Response: Thanks. We have removed the confusing grouping information.

L129 and 132: twice Figure 1, remove one **Response: Changed. Thanks.**

L133: “a generally high level of fitness”. “Generally” does not mean anything... statistically? So we expect a pvalue, a statistical test.

Response: A pvalue has been added. Thanks.

L139: all strains grew well at 37°C so the title of this part (the clinal strain displays a higher fitness at thigh temperature under submerged growth conditions) is false. I reckon it is true that the clinical strain grew better than others at 37°C, but this is not what it is written.

Response: Correct! Sorry for the mistake, which has been amended. Thanks.

L139-141: it is written that 37°C is the condition of the human bloodstream or of plant xylem vessels; I did not know that the plant xylem vessels were at that temperature??

Response: Changed. Thanks.

L145-151: why the ability to penetrate a cellophane was not tested at 37°C to mimic bloodstream?

Response: As stated in the same paragraph, we do not observe much growth at 37°C on solid medium; therefore, it is not possible to perform a penetration assay.

L153-154: Optical mapping is a method to joining scaffolds based on restriction enzymes, not to “investigate whether LS chromosomes are present”. This sentence

should be modified.

Response: Changed. Thanks.

L162/163: this sentence could be removed, redundant with what is written just below

Response: Changed. Thanks.

L164-169: should be in the Method, too detailed here.

Response: We have simplified this passage. Thanks.

L170: The supplementary file 3 is "PCR amplification of all PacC alleles in NRRL 32931". This has nothing to do here.

Response: Removed. Thanks.

L177: most sequences belonging

Response: Changed. Thanks.

L181: "core regions defined as scaffolds". "Scaffold" is a term already existing, and this is not its meaning.

Response: Changed. Thanks.

L192: transposons. Which category? Is a category statistically more represented than others?

Response: These transposons include DNA transposons. However, most of transposons do not have annotations.

L211: were **Response: Changed. Thanks.**

L208: LS genes were distinct. **Response: Changed. Thanks.**

L223-233: a discussion is missing here, it is just results, I don't get what the authors wanted to highlight here

Response: This is an excellent point. We have added a discussion in the revised manuscript accordingly.

L238: were **Response: Changed. Thanks.**

L234-240: again I don't get what the authors wanted to highlight here

Response: We have added a discussion of this in the revised manuscript accordingly. Thanks.

L241: lacked **Response: Changed. Thanks.**

L262: was **Response: Changed. Thanks.**

L264: a problem with the format of the bibliography **Response: Changed. Thanks.**

L265-266: do the authors suggest a role of transposable elements in the acquisition of genes in LS chromosomes? There are plenty of examples in the literature and this point could be discussed

Response: yes, you are correct.

L278-282: Horizontal transfer?

Response: Excellent point. We have added a short paragraph in the Conclusion section to address the genome dynamics and the contribution of transposable elements in this revised manuscript.

L344: Cylindrocarpon **Response: Changed. Thanks.**

L347-351: It could be interesting to cite the example of Candida auris, a new global emerging threat? **Response: Changed. Thanks.**

L356: which statistical test? Pvalue?

Response: Excellent point! We have added p-values.

L362: over is not needed here **Response:** Changed. Thanks.

L421: 28, 30 and 33 **Response:** Changed. Thanks.

L424: “similar” results. Which statistical test did the authors performed to conclude this?

Response: A Student’s *t*-test was used to analyze the data presented in Figure 2. We removed the ambiguous phrase from the Materials & Methods section.

L543: performed **Response:** Changed. Thanks.

L545: Sanger **Response:** Changed. Thanks.

Figure 1: The legend is not a figure legend, and details should be put into the Method section. For example, in a figure legend there is no need to have the name of the software used for aligning sequences or to know the number of steps for the most parsimonious tree. By the way, with molecular data it is more appropriate to perform ML analyses in which we have to select the best-fit model of DNA evolution (modeltest, Posada and Crandall 1998). I don’t know which model of DNA evolution was chosen for the ML analysis, it is not written in this legend (where it is not its place) nor in the method section (where it should be, with all details about the phylogenetic analysis). The scale of the tree should be in number of substitutions per site rather than in steps. It could be great to have the number of LS chromosomes in the different strains shown in that tree when information is available, or a figure on the right side representing these LS chromosomes in the different strains.

Response: A new ML phylogeny has been generated using a general time reversible model in MEGA (v7.0.20). Analysis details have been added in the Materials & Methods section.

Figure 2: A) the scale bar is not present in all pictures. It would be more easy for the reader to have the origin of the strain next to the name.

L981 and 983: “before” and “after”. I had to read twice to understand the sentences.

Response: Changed. Thanks.

Figure 3: L988 no ‘of’ between genome and Fol4287

Part B: why showing the pulse filed gel and its inverse image? I think only one is needed they are the same...

L1001: red in lowercase

Response: Changed. Thanks.

Figure 5: Part A: maybe the sequences of the clinical strain NRRL 32931 should be in another color to render the figure easy to understand. There is no information about how this phylogenetic tree was made (ML? bootstrap? Root?) All this should appear, and a detailed description should be added in the Method section. The legend does not specify what the right part of the figure 4A represents.

Part C: The sentence is unclear.

Part D: a statistical test to assume that the expression increase with the pH?

Part E: the legend is too short, it does not explain correctly the 4 images.

Response: Since we added Figure 4, Figure 5 is now Figure 6. As

suggested, a detailed description of the phylogeny construction was added in the Materials & Methods section. Also, we have revised the legend.

Figure 6: A (it is written in lowercase on the figure): why did the authors put the name of the gene (which is non-informative except if we would like to retrieve the sequence from the genome) for *Fusarium oxysporum* and the name of the species for all other nodes? There is no pink highlighted box as mentioned in the legend. As in figure 5, how was done the tree? Which method? A detailed description of the analysis should be added in the Method section.

***Response:* Since we added Figure 4, Figure 6 is now Figure 7. We have added gene names in Figure 6A. A detailed description of the analysis has been added in the Materials & Methods section.**

Figure 7: the legend of the color should not be separated in two different graphs. The legend is unclear, what was the experiment? For the Figure 2 the authors provided a lot of information, here nothing. A quick summary of the experiment is necessary for the understanding of the figure.

***Response:* Since we added Figure 4, Figure 7 is now Figure 8. We have changed the legend and added a summary of the experiment.**

Reviewer #3: In the paper “The genome of an opportunistic fungal pathogen *Fusarium oxysporum* carries a unique set of lineage-specific chromosomes”, the authors present their genome analysis of the pathogen *Fusarium oxysporum* from the blood of a patient with leukemia. *Fusarium oxysporum* is known for horizontally acquiring lineage-specific (LS) chromosomes. The authors of this manuscript also focus on the genes found in the LS chromosome – noting that some genes present appear to be correlated with pathogenicity, and state that LS chromosomes of clinical isolates have not been previously reported. The authors also assess growth rates of this and a small number of other isolates, and MIC for some antifungals for just this one isolate. They also present an RNAseq dataset that they use to gauge the completeness of their genome assembly, and look at the expression of a small group of transcription factor homologs (*pacC*).

Although the authors have done a reasonable amount of work characterising this isolate, the write-up is of a pretty low quality (see my comments for some examples). Figures are also mostly of a low quality (especially Fig 1, 3 and 4). Many of the analysis appear to be lacking a clear overview and presentation, in addition to pulling out the most interesting findings. The RNAseq experiment and the structural predictions appear to be particularly rushed and lacking key details. I’m concerned there has been no multiple correction for the GO-term enrichment, and a table of the uncorrected values discussed are not even presented in the manuscript – despite being one of the main results presented. MICs should also be taken for a panel of clinical and non-clinical *Fusarium* isolates to provide some overview of rang.

Response: Thanks for the careful review. A GO enrichment table has been added as Supplementary Table 8. *P*-values less than 0.05 are in black and greater than 0.05 are in grey.

Additional comments:

Page 4 line 79-80 “Fusariosis, an invasive fungal infection caused by *Fusarium* spp., is listed as the second most common opportunistic mold infection after aspergillosis 6,7”. The two citations are for papers on Fusariosis specifically, and not systematic reviews on the most common opportunistic mold infections as I believe is implied (i.e. listed). I do not see lists of most common infections in those papers, and it's dubious that Fusariosis, which is often described as rare, is more common than Candidiasis or Cryptococcosis for example. Please check or clarify this statement.

Response: Agreed. “The second most common” here pertains to opportunistic mold infections only.

Page 4 line 80: “positive blood cultures were detected among approximately 50% of fusariosis patients”. How were the other 50% of fusariosis patients diagnosed?

Response: Most clinical diagnoses are conducted in specialized facilities. Both a tissue sample from the infection site, and a blood sample from the patient were collected for microscopy inspection and for recovering the pathogen using culture methods. Fifty percent of patients diagnosed with fusariosis give positive results from blood culture. This is noteworthy, as blood cultures rarely provide positive results for other mycotic infections.

Page 4 line 82: “Because *Fusarium* spp. are broadly resistant to most clinically available antifungals [9]”. The citation is to a paper on fusariosis in a Brazilian hospital that does no characterisation of antifungals at all. The paper doesn’t even mention the word antifungal. Find a suitable reference for this statement or omit it.

Response: We apologize for the mistake. We have updated the citation.

Page 5 line 98: “To address these questions, we characterized *F. oxysporum*”. This is very vague. In what way did you characterise it? Also, this sentence at the start of a paragraph refers to the sentence in the previous paragraph, which are not phrased as questions.

Page 5 line 99: “We show here that this” needs rewording.

Response: After further analysis, this paragraph was reworded.

Page 5 line 106: Presenting a brief summary of the results at the end of the introduction is common, but for brevity, p-values of GO-term enrichment should probably be saved for the results. **Response:** Changed as suggested.

Page 5/6 lines 116-119: “The strain is available upon request ...” is not results. Move to methods. **Response:** Changed! Thanks.

Page 6 line 127: “Phylogenetically, NRRL32931”. The 1st and 2nd paragraph was about the isolate and its origin, which seem more suited to the methods if anything, and this paragraph seems to have skipped the sequencing step before launching into phylogenetics. A very short description of the sequencing and computational methods used (assuming they were) to generate a phylogenetic tree is required. Why has this been excluded?

Response: Excellent points. We have moved the analysis of the description to the Materials & Methods section and added a short description to support the phylogeny.

Page 6 line 133: “Phenotypically, ”. Delete. **Response:** Changed. Thanks.

Page 6 line 138: This paragraph is focused on the growth rates of isolates at different temperatures, and says that data is not shown for isolates at 34C. Why is not shown? Also what is Fo47? You say non-pathogenic, but where has it come from?

Response: We used a plate-based assay for all three strains. The results are now included as Supplementary Figure 1. Fo47 is a non-pathogenic strain widely used as a biocontrol agent. The reference was provided in the previous paragraph.

Page 7 line 164-165: “The genome assembly of NRRL 32931 was generated... and is available at NCBI (AFML01000000)”. On NCBI, AFML01000000 is listed as isolate 3-a, and the name “32931” does not appear. Can you confirm these are the same isolate, and if so explain the discrepancy in the names?

Response: Thanks for the careful review. “NRRL 32931” was the name from the ARS Culture Collection and 3-a was the name used for the sequencing

project. Although both names refer to the same strain, we agree this may cause confusion and have contacted NCBI to add NRRL 32931 to the project name. NCBI agreed to this change and is in the process of modifying all related files in the NCBI database.

Page 7 line 175: “By contrast, mapping of the assembled sequence to the LS chromosomes from 12 to 15 only resulted in 31.1%, 42.5%, 18.5% and 26.4% mapping rates, respectively” needs rewording for clarity. Also, what is the take home message of this?

Response: The relatively low mapping rates are likely due to the high proportion of repeats in these chromosomes. We have included a sentence that states our take home message. “Due to the high proportion of repeats present, LS chromosomes were highly fragmented. Consequently, mapping of the assembled sequence to the LS chromosomes from 12 to 15 only resulted in 31.1%, 42.5%, 18.5%, and 26.4% mapping rates, respectively.”

Page 11 line 214: “Interestingly, “. Delete.

Response: Deleted. Thanks.

Page 12 lines 262-268: Needs rewording and restructuring. Also are you discussing GO-terms? Because it is listed as only “functional category” which could be any number of things. Also, do these copper-binding proteins have GO terms relating to iron homeostasis? It might be helpful to list all of the “functional category’s” that this gene has assigned to it (rather than just the one) to give an indication of its possible function.

Response: We have added a full list of GO terms for top 10 enriched GO terms of NRRL 32931 and FoI4287 in Supplementary Table 8.

Page 13 line 301-302: “we noticed ... is distantly related”. Be more specific.

Response: We have reworded this sentence: “Indeed, FOYG_17127 is related to copper-binding laccases, which were reported as virulence factors in another human pathogenic fungus *Cryptococcus neoformans*.”

Page 16 line 361: “An opportunistic infection occurs when the host immunity is impaired.”. No. An opportunistic infection does not always occur when the host immunity is impaired. Also, opportunistic infections have a wide range of causes – not only impaired immunity.

Response: The sentence was corrected to “An opportunistic fungal infection is associated with a patient with impaired immunity.”

Page 20 line 463: Provide citations or NCBI accessions for these RNAseq datasets.

Response: Four Illumina Hiseq runs were generated to assist in genome annotation as part of the Fusarium comparative genomics project. The accession numbers have been added to the manuscript (SRX025824, SRX025823, SRX026545, SRX027736).

Page 20 line 474: “circumvented a major memory problem”. Badly described.

Response: Changed to “resolved a major memory problem”. Thanks.

Page 21 line 495: You didn't use multiple correction with your Fisher exact test?

Response: A Benjamin and Hochberg FDR correction with a cutoff of 0.05 was also applied. We clarified this in the text.

Figure 1. Is this ML or MP? Can you explain “250 steps” as a scale bar?

Response: The phylogeny in Figure1 is inferred from a maximum likelihood analysis. In this phylogeny, we have used number of substitutions per site instead of steps. Details have been added to the figure legend.

Reviewers' comments:

Reviewer #2 (Remarks to the Author):

I reviewed this manuscript the first time and I reckon this version is more complete and explore very well the LS chromosomes in a clinical isolate of *Fusarium oxysporum*. However, I still have some concerns:

First, the authors added in this new version another human pathogenic isolate (NRRL 47514), which is great. Indeed, it gives more power to their conclusions. However, they don't change some parts of the manuscript and it renders the reading unclear sometimes (do this manuscript describe one or two clinical isolates?). For example, in the abstract it is just mentioned that the authors compared three strains (one from human, one from tomato and one nonpathogenic). Also, titles in the Results and Discussion section always mention the NRRL 32931 strain, never the NRRL 47514 strain, because most of the time this new strain was not added in the experiments or analyses (e.g. phenotypic experiments). Importantly, this new strain is not in the phylogenetic tree (Figure 1): it would be great to have it in the tree, it is indeed important to know its phylogenetic relationships with other strains ; are the two human pathogenic strains cluster together? Also it is written that the NRRL 47514 and NRRL 32931 isolates shared some parts of LS sequences (about a third). But it is unclear from the reading if the NRRL 47514 strain carries other LS chromosomes not present in the NRRL 32931? How many/what proportion? It looks like the authors did a one way comparison, it would be nice to analyze the LS chromosomes in the two strains similarly, which should be feasible as the strain NRRL 47514 was sequenced using Illumina and Pacbio sequencing.

Second, the authors performed phenotypic experiments and did some statistical tests using three strains... This leads to this kind of sentences L131: "NRRL 32931 displayed a statistically higher level of fitness at elevated temperatures than FoI287 and Fo47". The comparison of three strains is not enough to make any conclusion of "adaptive traits", to do any conclusion at all and to perform any statistical tests... all this part (L112-123 and 130-146 and sentences in the introduction, conclusion) have to be removed from the manuscript, or experiments have to be redone with an appropriate sampling.

Third, the story about paralogs of *pacC* is incomplete as it is (L235-276). The authors made a phylogenetic tree (figure 6A) but did not discuss it: how do the authors explain this topology? For me, it says that there was a first duplication event in the common ancestor *PacC_O* and then others, given rise to 3 *PacC* paralogs.

L249: "the presence of TEs was also observed for the LS *pacC* paralogs". What does that mean? For me, it means that there are some TEs are within these paralogs. However, if I read next sentences, this is not what the authors meant and what the figure 6C shows. Importantly, the authors wrote "the nucleotide sequence identity among this expanded *pacC* gene family is below 90%, much lower than that of orthologous genes within the FOOSC, which supports the horizontal origin of this group of genes". Why? I really don't get why the authors suggest this hypothesis and I am not agree with this statement, in the tree (figure 6A) there is no other species than *Fusarium* but only the two human pathogenic strains.

Other minor comments:

L59: "clinical isolates" but in the abstract it is only one.

L96: now two isolates, it has to be modified.

L98-99: All this part has to be removed from the manuscript, or experiments have to be redone with an appropriate sampling (more than one human pathogenic strain, more than one phytopathogenic strain and more than one nonpathogenic strain).

L101-104: we have to understand since the beginning of the reading that two human pathogenic isolates have been studied in that manuscript. This sentence would be better placed at the beginning of the paragraph, L97.

L148-158: what about the other human pathogenic strain?

L189: again, what about the other human pathogenic strain? It is clear these parts were written without this additional strain. Which transposons? The authors named the one found in the phytopathogenic strain but not the ones present in the human pathogenic one..

L212: "NRRL 32931 and NRRL 47514"

L290: orthologs of what? Better to rewrite it here

L290-301: do the authors think the horizontal transmission occurred in humans as this gene is present in human pathogens? Is it frequent to have *Exophiala oligosperma*, *E. cenobiotica*, *Aspergillus calidoustus* and *Fusarium oxysporum* present simultaneously in the human body? Maybe one or two sequences could be added on that? And who is the donor? It would be better to put the entire names in the phylogenetic tree, it will help to understand better at a first glance the figure 7.

L341: NRRL 47514 should also be tested for susceptibility to antifungals. How many strains were tested? This information is missing

L345: how many clinical strains?

L384: a little bit too overinterpret to talk about "host-specialize adaptation" with only one strain from a mammalian host...

Conclusion: the conclusion should include the strain NRRL 47514.

Method: Phylogenetic analyses: no bootstrap support? But in the figure 1 there are some bootstraps. It is not mentioned in the mat and meth, please add it.

L539: Why GTR?

L607: Part Antifungal susceptibility: it is not written how many strains were included in that experiment.

Figures and figure legends

Figure 1: "rooted on the sequences of *F. verticillioides*". I can only see one sequence of *F. verticillioides* in the tree, there should be two. Nothing is said about bootstraps in the mat and meth, it has to be added in the section "phylogenetic analyses".

Figure 2: statistics have been made using 3 strains! this figure should be removed, and all things related to that in the manuscript too. Or experiments have to be redone with a more representative sampling.

Figure 3B: it would be great to add what columns 1 to 4 mean directly on the figure.

Figure 5: I don't really get the point of this figure. Just writing the enrichment of proteins with metal ion binding functions in the manuscript is enough. Moreover, it could be misleading as there are two circles for "binding" and "metal ion binding" but given the structure of the gene ontology terms, the latter is included in the first one.

Figure 8: in the legend, there is no information about sampling: how many strains of *A. fumigatus* and *F. oxysporum*?

Reviewer's Responses to Questions

I reviewed this manuscript the first time and I reckon this version is more complete and explore very well the LS chromosomes in a clinical isolate of *Fusarium oxysporum*.

Response: We appreciate this recognition.

However, I still have some concerns:

First, the authors added in this new version another human pathogenic isolate (NRRL 47514), which is great. Indeed, it gives more power to their conclusions. However, they don't change some parts of the manuscript and it renders the reading unclear sometimes (do this manuscript describe one or two clinical isolates?). For example, in the abstract it is just mentioned that the authors compared three strains (one from human, one from tomato and one nonpathogenic). Also, titles in the Results and Discussion section always mention the NRRL 32931 strain, never the NRRL 47514 strain, because most of the time this new strain was not added in the experiments or analyses (e.g. phenotypic experiments). Importantly, this new strain is not in the phylogenetic tree (Figure 1): it would be great to have it in the tree, it is indeed important to know its phylogenetic relationships with other strains ; are the two human pathogenic strains cluster together? Also it is written that the NRRL 47514 and NRRL 32931 isolates shared some parts of LS sequences (about a third). But it is unclear from the reading if the NRRL 47514 strain carries other LS chromosomes not present in the NRRL 32931? How many/what proportion? It looks like the authors did a one way comparison, it would be nice to analyze the LS chromosomes in the two strains similarly, which should be feasible as the strain NRRL 47514 was sequenced using Illumina and Pacbio sequencing.

Response: Thanks for pointing this out! We have carefully revised manuscript accordingly. Specifically, we have:

- 1) removed phenotypic experiments from the current version of the manuscript.
- 2) added the keratitis strain NRRL 47514 in the phylogeny (Figure1) and discussed their relatedness.
- 3) conducted pairwise comparisons of among three genomes, which is presented in a revised version of Figure 3.

In summary, 37% NRRL 32931 masked LS sequences have homologous sequences in the NRRL 47514 genome, while only 2.3% in the Fol4287 genome. With larger LS region in the NRRL 47514 genome, two human pathogen shared LS sequences represent 17% of NRRL 47514 masked LS sequences, about 7% of NRRL 47514 masked LS sequences also have homologous sequences in the genome of Fol4287.

Second, the authors performed phenotypic experiments and did some statistical tests using three strains... This leads to this kind of sentences L131: "NRRL 32931 displayed a statistically higher level of fitness at elevated temperatures than Fol287 and Fo47". The comparison of three strains is not enough to make any conclusion of "adaptive

traits”, to do any conclusion at all and to perform any statistical tests... all this part (L112-123 and 130-146 and sentences in the introduction, conclusion) have to be removed from the manuscript, or experiments have to be redone with an appropriate sampling.

Response: Thanks for your suggestion. We have removed this section from the revised manuscript.

Third, the story about paralogs of *pacC* is incomplete as it is (L235-276). The authors made a phylogenetic tree (figure 6A) but did not discuss it: how do the authors explain this topology? For me, it says that there was a first duplication event in the common ancestor *PacC_O* and then others, given rise to 3 *PcaC* paralogs.

L249: “the presence of TEs was also observed for the LS *pacC* paralogs”. What does that mean? For me, it means that there are some TEs are within these paralogs. However, if I read next sentences, this is not what the authors meant and what the figure 6C shows. Importantly, the authors wrote “the nucleotide sequence identity among this expanded *pacC* gene family is below 90%, much lower than that of orthologous genes within the FOSC, which supports the horizontal origin of this group of genes”. Why? I really don’t get why the authors suggest this hypothesis and I am not agree with this statement, in the tree (figure 6A) there is no other species than *Fusarium* but only the two human pathogenic strains.

Response: Since we have removed Figure 2 and Figure 5, Figure 6 is now Figure 4.

- 1) We have added a sentence to explain the topology of the phylogeny.
- 2) We agree the description on TE is not clear and we have changed the sentence to be “The presence of transposable elements was also observed in the flanking regions of the LS *pacC* paralogs.”
- 3) Also, we changed the sentence in L253 to “The nucleotide sequence identity among this expanded *pacC* gene family is below 90%, much lower than that of orthologous genes within the FOSC (~99%), which may result from horizontal origin of this group of genes or rapid sequence divergence after duplication”.

Other minor comments:

L59: “clinical isolates” but in the abstract it is only one.

Response: Thanks for pointing this out. We have added the keratitis strain NRRL 47514 in the abstract. We also modified the Author summary accordingly.

L96: now two isolates, it has to be modified.

Response: We have revised this sentence as the following:
In this study, we analyzed two *F. oxysporum* human isolates, NRRL 32931: a strain isolated from the blood of a leukemia patient with invasive fusariosis; and NRRL 47514 (MRL 8996), a strain isolated from a contact lens associated with the USA 2005/06 *fusarium* keratitis outbreak.

L98-99: All this part has to be removed from the manuscript, or experiments have to be redone with an appropriate sampling (more than one human pathogenic strain, more than one phytopathogenic strain and more than one nonpathogenic strain).

Response: We have revised the manuscript and removed phenotypic experiments accordingly.

L101-104: we have to understand since the beginning of the reading that two human pathogenic isolates have been studied in that manuscript. This sentence would be better placed at the beginning of the paragraph, L97.

Response: As suggested, we have integrated L101-104 into the first sentence of this paragraph (L67-70) in revised manuscript.

L148-158: what about the other human pathogenic strain?

Response: We have changed the title of this section to “Comparative genomics reveals structural compartmentalization in two human pathogenic *F. oxysporum* genomes”. We added NRRL 47514 genome in a separate paragraph.

L189: again, what about the other human pathogenic strain? It is clear these parts were written without this additional strain. Which transposons? The authors named the one found in the phytopathogenic strain but not the ones present in the human pathogenic one..

Response: We have added NRRL 47514 in transposon analysis. Both NRRL 32931 and NRRL 47514 are lack of the transposons associated with pathogenicity in some phytopathogenic strains, such as MIMPs and Helitrons. The transposons without name in human strains are because we are not able to name them based on their structure.

L212: “NRRL 32931 and NRRL 47514”

Response: Changed. Thanks.

L290: orthologs of what? Better to rewrite it here

Response: We have revised this sentence to “Moreover, we identified PC homologs in three other opportunistic fungal pathogens, including *Exophiala xenobiotica* (XP_013312618.1) and *Exophiala oligosperma* (XP_016256162.1), which are black yeasts of the Herpotrichiellaceae family that includes diverse human and other vertebrate pathogens; and *Aspergillus calidoustus* (CEL09498.1), a causal agent of invasive aspergillosis”.

L290-301: do the authors think the horizontal transmission occurred in humans as this gene is present in human pathogens? Is it frequent to have *Exophiala oligosperma*, *E. xenobiotica*, *Aspergillus calidoustus* and *Fusarium oxysporum* present simultaneously in the human body? Maybe one or two sequences could be added on that? And who is the

donor? It would be better to put the entire names in the phylogenetic tree, it will help to understand better at a first glance the figure 7.

Response: So far, only six fungal species genomes have CP homolog, which are all included in our study. Right now, we do not know who the donor is. But we can see CP in *F. oxysporum* is not vertically inherited from its ancestry. Since we have removed Figure 2 and Figure 5, Figure 7 is now Figure 5. We have put the entire species names in the phylogeny of Figure 5.

L341: NRRL 47514 should also be tested for susceptibility to antifungals. How many strains were tested? This information is missing

L345: how many clinical strains?

Response: it is unfortunate that we are not able to add NRRL 47514 in the susceptibility tests (done by a person who is no longer with us). Since the antifungal resistance was reported in multiple papers, we feel this information is not crucial and decided to remove the figure all together.

L384: a little bit too overinterpret to talk about “host-specialize adaptation” with only one strain from a mammalian host...

Response: We have revised our language to niche adaptation.

Conclusion: the conclusion should include the strain NRRL 47514.

Response: We have revised this paragraph with NRRL 47514.

Method: Phylogenetic analyses: no bootstrap support? But in the figure 1 there are some bootstraps. It is not mentioned in the mat and meth, please add it.

Response: We have revised this paragraph with NRRL 47514. Also phylogenetic analysis with bootstrap test is added in the Materials & Methods section.

L539: Why GTR?

Response: The phylogeny generated from GTR has the highest likelihood.

L607: Part Antifungal susceptibility: it is not written how many strains were included in that experiment.

Response: Since the antifungal resistance was reported in multiple papers, we feel this information is not crucial and decided to remove the figure all together.

Figures and figure legends

Figure 1: “rooted on the sequences of *F. verticillioses*”. I can only see one sequence of *F. verticillioses* in the tree, there should be two. Nothing is said about bootstraps in the mat and meth, it has to be added in the section “phylogenetic analyses”.

Response: Bootstrap test is added in the Materials & Methods section.

Figure 2: statistics have been made using 3 strains! this figure should be removed, and all things related to that in the manuscript too. Or experiments have to be redone with a more representative sampling.

***Response:* As suggested, we have removed this figure.**

Figure 3B: it would be great to add what columns 1 to 4 mean directly on the figure.

***Response:* Since we removed Figure 2, Figure 3 is now Figure2. We have added column names in the figure.**

Figure 5: I don't really get the point of this figure. Just writing the enrichment of proteins with metal ion binding functions in the manuscript is enough. Moreover, it could be misleading as there are two circles for "binding" and "metal ion binding" but given the structure of the gene ontology terms, the latter is included in the first one.

***Response:* As suggested, we have removed this figure.**

Figure 8: in the legend, there is no information about sampling: how many strains of *A. fumigatus* and *F. oxysporum*?

***Response:* See above, we have removed this figure.**